# Nuclear speed and cycle length co-vary with local density during syncytial blastoderm formation in a cricket

Seth Donoughe [1,6]✉, Jordan Hoffmann[2], Taro Nakamura [1,7], Chris H. Rycroft [2,3]✉ & Cassandra G. Extavour [1,4,5]✉

The blastoderm is a broadly conserved stage of early animal development, wherein cells form a layer at the embryo's periphery. The cellular behaviors underlying blastoderm formation are varied and poorly understood. In most insects, the pre-blastoderm embryo is a syncytium: nuclei divide and move throughout the shared cytoplasm, ultimately reaching the cortex. In *Drosophila melanogaster*, some early nuclear movements result from pulsed cytoplasmic flows that are coupled to synchronous divisions. Here, we show that the cricket *Gryllus bimaculatus* has a different solution to the problem of creating a blastoderm. We quantified nuclear dynamics during blastoderm formation in *G. bimaculatus* embryos, finding that: (1) cytoplasmic flows are unimportant for nuclear movement, and (2) division cycles, nuclear speeds, and the directions of nuclear movement are not synchronized, instead being heterogeneous in space and time. Moreover, nuclear divisions and movements co-vary with local nuclear density. We show that several previously proposed models for nuclear movements in *D. melanogaster* cannot explain the dynamics of *G. bimaculatus* nuclei. We introduce a geometric model based on asymmetric pulling forces on nuclei, which recapitulates the patterns of nuclear speeds and orientations of both unperturbed *G. bimaculatus* embryos, and of embryos physically manipulated to have atypical nuclear densities.

[1] Department of Organismic and Evolutionary Biology, Harvard University, Cambridge, MA, USA. [2] John A. Paulson School of Engineering and Applied Sciences, Harvard University, Cambridge, MA, USA. [3] Computational Research Division, Lawrence Berkeley Laboratory, Berkeley, CA, USA. [4] Department of Molecular and Cellular Biology, Harvard University, Cambridge, MA, USA. [5] Howard Hughes Medical Institute, Chevy Chase, MD, USA. [6] Present address: Department of Molecular Genetics and Cell Biology, University of Chicago, Chicago, IL, USA. [7] Present address: Division of Evolutionary Development, National Institute for Basic Biology, Okazaki, Japan. ✉email: donoughe@uchicago.edu; chr@math.wisc.edu; extavour@oeb.harvard.edu

Proper positioning of nuclei is essential for cellular function[1,2]. The task of correctly positioning nuclei is further specialized in syncytial cells—those with multiple nuclei sharing the same cytoplasm[3,4]. Naturally occurring syncytia include animal muscle cells, heterokaryotic fungi, plant endosperm, and early cleavage stage arthropod embryos[5–8]. Among arthropods, there have likely been multiple independent evolutionary origins of a syncytial phase of embryonic development[8]. Here, we focus on insects.

When an insect egg is fertilized, the oocyte and sperm pronuclei fuse, forming the zygotic nucleus within a single, large cell[9,10]. In most insect taxa there follows a series of syncytial cleavages—nuclear divisions without cytokinesis[9–11]. As the divisions proceed, nuclei move throughout the cytoplasm of the embryo. Although some nuclei remain submerged in the middle of the embryo, most of them travel into the periplasm, a region of cytoplasm at the periphery of the embryo[9,10]. The nuclei in the periplasm comprise a syncytial blastoderm, a single layer of nuclei surrounding the cytoplasm in the interior[9,10].

Syncytial blastoderm formation has been studied most closely in the fruit fly *Drosophila melanogaster*[12,13]. After fertilization, *D. melanogaster* undergoes 13 synchronous divisions[12]. During cycles 4 through 6, nuclei spread out along the anterior-posterior (A-P) axis without entering the periplasm (a process referred to as "axial expansion")[13–16]. Nuclear movements along the A-P axis appear to be caused by the contraction of a subset of the cortex, which generates a cytoplasmic flow that carries nuclei towards the poles[17,18]. It has also been suggested that local forces act on nuclei via their astral microtubules (MTs) pulling the nuclei toward the adjacent F-actin network[19] and/or mutual repulsion among neighboring nuclei[20]. Then, during cycles 7 through 9, the nuclei simultaneously move into the periplasm (leaving a small subset behind as "yolk nuclei," also called "vitellophages")[9,20,21]. Finally, during cycles 10 through 13, the nuclei remain in the periplasm, arranged as a single layer[15,21,22]. They increase in local nuclear density and assume an orderly geometric spacing[23–28].

Among insects with syncytially cleaving embryos, it appears to be universal that nuclei travel through the interior cytoplasm and into the periplasm[9,10,29], yet species differ dramatically with respect to the timing, speeds, and the paths that their nuclei traverse while getting there[9,10,29]. This raises the question of how different insect species generate such embryological diversity. There is evidence from fixed preparations that some of the mechanisms described in *D. melanogaster*—namely cytoplasmic flows and MT-mediated pulling—might be operating in more distantly related insects[30,31]. To assess such possibilities, quantitative, nucleus-level data on the dynamics of syncytial blastoderm formation are needed for species other than *D. melanogaster*. Therefore, we set out to investigate an informative comparator: the two-spotted field cricket *Gryllus bimaculatus* (order: Orthoptera).

*G. bimaculatus* is a powerful complement to *D. melanogaster* for the study of syncytial development. *G. bimaculatus* embryos are larger (approximately five-fold longer and three-fold wider)[32,33] and their blastoderm formation occurs more slowly (14 hours at 28.5 °C versus 3 hours at 25 °C)[22,34]. *G. bimaculatus* is hemimetabolous, and its embryonic development differs in many respects from that of *D. melanogaster* and other model holometabolous insect species[33]. It likely retains many features of ancestral insect embryogenesis, unlike the relatively derived fruit fly model[33]. Crucially, a transgenic line of *G. bimaculatus* expressing a constitutive ubiquitous Histone2B-Enhanced Green Fluorescent Protein (H2B-EGFP) fusion has been generated[34]. The fusion protein presents a strong fluorescent contrast between syncytial nuclei and the surrounding cytoplasm during the entirety of pre-blastoderm development[34]. This enabled us to record, track, and analyze the movements of nuclei during syncytial development, starting from mitotic cycle ~2-4 and ending at the formation of the blastoderm.

We used multiview lightsheet and confocal microscopy to capture three-dimensional timelapse (3D + T) datasets and epifluorescence microscopy to capture two-dimensional timelapse (2D + T) datasets of syncytial development. We used a semi-automated approach to reconstruct nuclear tracks through space[35,36], and analyzed nuclear divisions, speeds, and movement orientations. We show that each of these nuclear behaviors co-varies predictably with local nuclear density rather than with axial position, lineage, or developmental timing. We also show that the patterns of nuclear migration are more consistent with active movement through the cytoplasm, rather than with passive movement resulting from being carried along in a cytoplasmic flow. Based on our empirical description, and inspired by previously published work on active nuclear migration in other contexts[19,37–40], we built a simple computational model of nuclear movement based on asymmetric pulling forces and local interactions among nuclei. This model recapitulates the main features of *G. bimaculatus* nuclear divisions, speeds, and orientations during syncytial development. We tested the model by experimentally altering nuclear density, finding support for the hypothesis that a locally acting mechanism causes nuclear speed and density to co-vary predictably. Finally, we use this model to generate falsifiable hypotheses about blastoderm formation in other insect species.

## Results

The *G. bimaculatus* syncytial blastoderm forms during approximately eight hours of development at 28.5 °C, followed by cellularization and coalescence of the embryonic rudiment[33,34] (Fig. 1a). We recorded *G. bimaculatus* syncytial blastoderm formation *in toto* by using 3D + T lightsheet microscopy to image H2B-EGFP transgenic embryos. We tracked the movements and divisions of nuclei as they expanded throughout the embryo, which enabled us to reconstruct nuclear lineages (Fig. 1b) and quantify nuclear density, speed, and direction of movement. Similar to *D. melanogaster*[13,15,16], some nuclei appear to move in a highly directed manner toward the poles of the embryo (Fig. 1b, arrowhead). Unlike *D. melanogaster*, however, *G. bimaculatus* nuclei move into the periplasm asynchronously, with some reaching the periplasm 240 minutes after those that first reach it (Fig. 1b).

**Nuclei are not predominantly moved by cytoplasmic flows.** We wanted to determine whether nuclei are moved passively by being carried along in a cytoplasmic flow[17,18], or instead moved by a local active force[19,20,31,38]. We generated a new stable transgenic line of *G. bimaculatus* that expressed myristoylated and palmitoylated mTomato (mtdT) protein in the embryo. The mtdT protein is predominantly localized to the cytoplasm in the immediate vicinity of each syncytial nucleus (i.e. within the "energid"[8,9]). It was excluded from putative yolk granules (Fig. 1c, white arrowheads; Supplementary Movie 1), but allowed us to see their outlines and follow their movements. We generated embryos that co-expressed mtdT and H2B-EGFP, which enabled us to image yolk granules and nuclear movements together (Fig. 1c; Supplementary Movie 1). We found that yolk granules and adjacent nuclei do not tend to move in the same direction together, even when they are quite close in space (e.g. within 40 μm, as shown in Fig. 1c and Supplementary Movie 1). This is inconsistent with a cytoplasmic flow that moves yolk granules and nuclei together. To further test the possibility of movement via cytoplasmic flows, we computed instantaneous pairwise

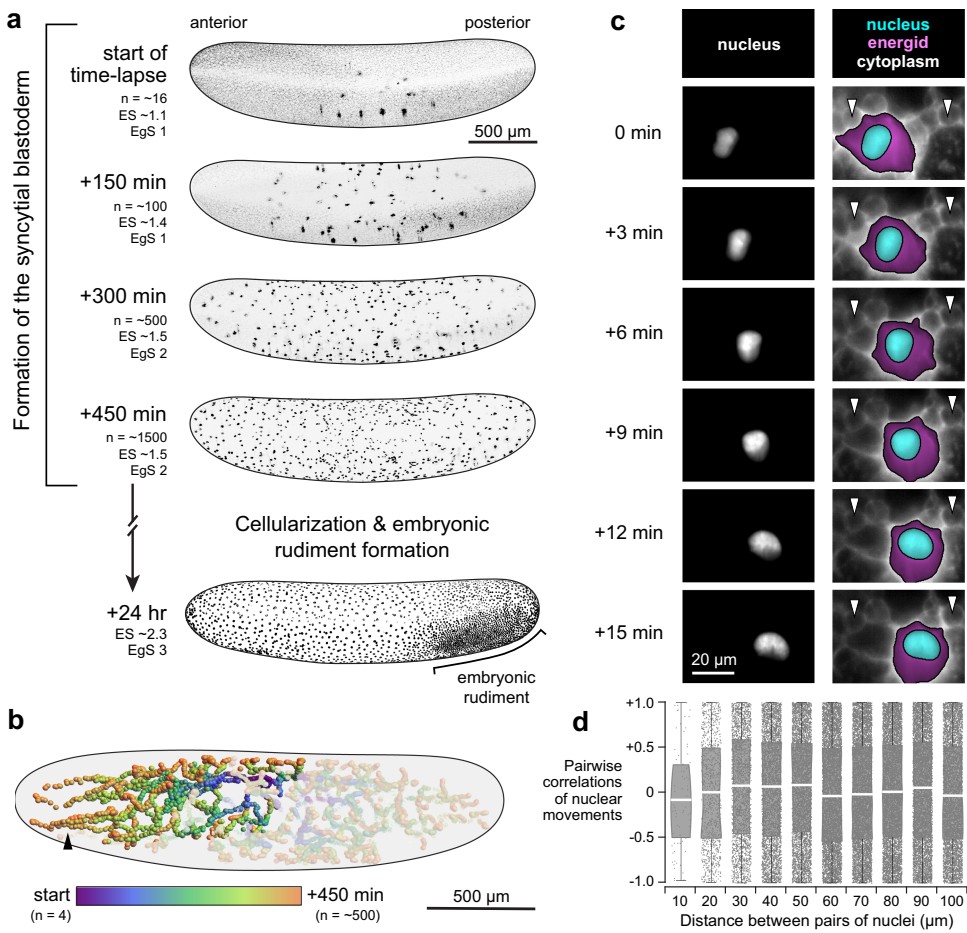

**Fig. 1 Overview of *G. bimaculatus* blastoderm formation and evidence that nuclei move actively, rather than being moved passively by flowing cytoplasm. a** Time points from the embryonic syncytial development of *G. bimaculatus*, displayed as *z*-projections of 3D stacks. Micrographs are from an H2B-EGFP transgenic *G. bimaculatus* embryo live-imaged using a lightsheet microscope over eight hours of development at 28.5 °C, capturing nuclear divisions and movements throughout the syncytial embryo. The nuclei (n) arrange into a single layer, after which cellularization occurs and the embryonic rudiment forms. Embryos are oriented laterally with ventral to the bottom and anterior to the left. Anterior is to the left in all subsequent figures. Embryonic stage (ES) and egg stage (EgS) are indicated for each time point[33]. **b** Nuclei were tracked to produce a 3D + T dataset of nuclear lineages. All nuclear tracks are displayed for an example embryo, with the lineage descended from a single nucleus highlighted. Color scale represents time; *n* = number of nuclei. Black arrowhead highlights nuclei that move in a highly directed manner toward the anterior pole. **c** Example time points from a 3D + T dataset of a transgenic *G. bimaculatus* embryo with nuclei and cytoplasm fluorescently marked (further details in Methods). Left column shows the nucleus channel, and the right column shows the cytoplasm channel, with the energid cytoplasm highlighted in magenta and the nucleus highlighted in cyan. White arrowheads mark two putative yolk granules that remain in place as the nucleus moves past them. **d** Pairwise correlations between the instantaneous movement vectors of pairs of non-sister nuclei. White line indicates median, box indicates 25th–75th percentiles, whiskers show range. Source data are provided as a Source Data file.

correlations between the movement vectors of all pairs of nuclei. If nuclei were embedded in a cytoplasmic flow, we would expect nearby nuclei to move more similarly to one another than nuclei that were farther apart. This was not the case. Irrespective of pairwise distance, pairs of nuclei exhibit a random pattern of movement correlations (overall correlation of pairwise values with separation distance yields $R^2 = 0.032$; Fig. 1d; this calculation, and those presented in Fig. 2 and Fig. 3, was done on a dataset with 310,741 nucleus-timepoints). These observations suggest that *G. bimaculatus* syncytial nuclei are not predominantly moved by cytoplasmic flows.

**Mitotic cycle duration is positively associated with local nuclear density.** Previous work showed that syncytial nucleus divisions in *D. melanogaster* are synchronous, and that divisions are coupled to the nuclear movements that underlie blastoderm formation[17,18]. We asked whether a similar mechanism operates

in *G. bimaculatus* embryos. For a coarse-grained measurement of proliferation, we calculated the percent change in the number of detectable nuclei over time (Fig. 2a). This metric displays a dynamic series of peaks and valleys, with each peak representing a pulse of divisions. These peaks occur initially at mean intervals of 49 minutes, and the time between peaks increases as development proceeds. This indicates that the cell cycle duration is approximately 49 minutes, which is a much greater time interval than the 8 to 9 minute cell cycle duration in a *D. melanogaster* embryo[12]. The sharpness of the peaks attenuates over time, indicating a decrease in relative synchronicity (Fig. 2a). Next, we sought to determine whether the mitotic cycles are collectively going out of phase and/or the mitotic cycle lengths are themselves changing. We measured the cycle length of the nuclei, finding that it changes markedly over the course of blastoderm formation. An example lineage is shown in Fig. 2b. During four successive cycles, mean cycle length increases from 49 to 87 minutes (Fig. 2b). There is also considerable variation in cycle length

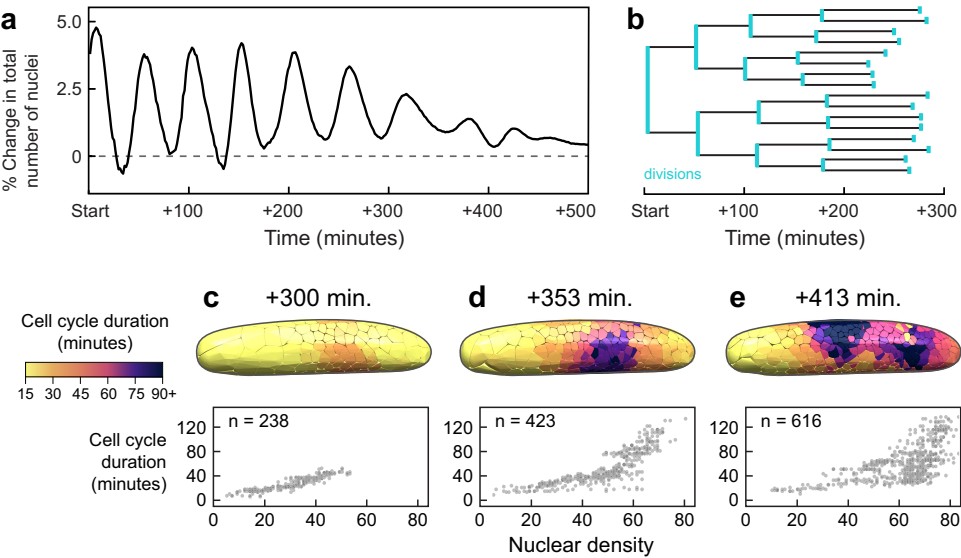

**Fig. 2 Mitotic cycle duration covaries with nuclear density, explaining a marked decline in division synchronicity. a** Percent change in the total number of observable nuclei over time. Divisions become increasingly asynchronous over the course of syncytial development. **b** Example lineage of dividing nuclei, starting with one of the nuclei at the four nucleus stage with divisions marked in turquoise. Over four division cycles, mean cycle duration increased from 49 to 87 minutes. **c–e** Cell cycle duration was positively associated with local density. Cell cycle duration was calculated in the vicinity of each nucleus by measuring the time elapsed until the number of nuclei within a 150 μm radius increased by 25%. Top row shows three example time points, with local cell cycle duration displayed as colored volumes, each of which contains a single nucleus. Bottom row shows scatterplots of the local cell cycle duration times and nuclear densities at each time point; n = number of nuclei. Source data are provided as a Source Data file.

among nuclei at the same cycle number (e.g. at cycle 4, SD of cycle length = 11 minutes; Fig. 2b). This is again in contrast to *D. melanogaster* embryos, in which cycle length increases over time, but all nuclei within a cycle have the same period[21,41]. We hypothesized that in *G. bimaculatus*, variation in local nuclear density gives rise to the heterogeneity of cycle length. To test this, we computed the local nuclear density and proliferation time in the vicinity of each nucleus throughout blastoderm formation (three example time points are shown in Fig. 2c–e). Within each time point, we also took all possible nucleus pairs and computed the percentage of instances where the nucleus with the higher local density also had a longer proliferation time. These percentages are: $t = 300$ minutes: 91%, $t = 353$ minutes: 88%, $t = 413$ minutes: 75%. We concluded that there is a positive association between local nuclear density and mitotic cycle length across a range of local densities and throughout syncytial blastoderm formation.

**Nuclear speed is biphasic and negatively associated with density**. Given that a single variable—density—helped to explain both spatial and temporal variation in mitotic cycle length, we wondered whether the speeds of nuclear movements also followed a similar coherent pattern. We calculated instantaneous speed of all nuclei and then plotted those speeds over time ($y$) vs. position along the A-P axis ($x$) (Fig. 3a). Nuclear speed oscillates back and forth between "fast" and "slow" (> 4 to < 1 μm per minute, respectively). It appeared that in the central region of the embryo, farthest from the anterior and posterior poles, peak speeds decrease earlier, and the speed oscillations dissipate sooner than at either pole (Fig. 3a, middle of A-P axis, t > 150 minutes). To illustrate this further, we ordered the nuclei according to their position along the A-P axis and then partitioned them into three terciles, with equal numbers of nuclei in each one (labeled "anterior", "middle", and "posterior" terciles; Fig. 3b). When we plotted speed over time for the nuclei contained in each tercile, we observed that nuclear speed oscillates for all three terciles. In the middle tercile, however, where density is higher than in the

other terciles at every developmental time point examined, the oscillation is less pronounced and diminishes sooner, than in the anterior and posterior terciles (Fig. 3c). We also noticed that the speed oscillations are qualitatively similar to the oscillations of the percent change in total number of nuclei (compare Figs. 2a and 3c; $R^2 = 0.976$ for the correlation of the first six peaks from each dataset). Therefore, we hypothesized that each nucleus's movements jointly depend on its local density and on the amount of time spent executing the cell cycle.

To test this, we computed time-since-last-division for each nucleus within the 200 minutes of development depicted in Fig. 3a. We divided the nuclei into three subsets according to their local nuclear density: "low," "medium," and "high" (<11, ≥11 and <29, ≥29 density units, defined here as the number of nuclei within a 150 μm radius). We plotted instantaneous nuclear speed for each density bin, with time re-zeroed to begin at the most recent division of each nucleus. This revealed density-associated speed oscillations for all nuclei, regardless of chronological age or spatial location within the egg (Fig. 3d). Nuclear speed alternated between relatively fast and slow phases, which we refer to as Phase A and Phase B, respectively. During Phase A, which is immediately after a division, each daughter nucleus moves relatively quickly (median Phase A speed: 2.3 μm per minute) for between 20 and 28 minutes. During Phase B, the nucleus remains largely stationary (median Phase B speed: 0.4 μm per minute) for between 10 and 20 minutes before again dividing and repeating this process (Fig. 3d). We plotted nuclear speed ($y$) vs. nuclear density ($x$) for Phase A and Phase B (Fig. 3e, f), and considering that both speed and density are by definition positive, in each case we fitted an exponential curve with the form $y = y_0 e^{-x/x_0}$. This yields a "density scale" ($x_0$) that captures how large of a change in density produces a given change in speed. The density scale of Phase A is 30.4 density units (90% CI: 28.8 to 32.1). For Phase B it is 120.4 density units (90% CI: 73.4 to 167.4). In sum, we found that Phase A nuclear movements have overall higher speeds that are strongly associated with density, while Phase B nuclear movements have lower speeds that are weakly

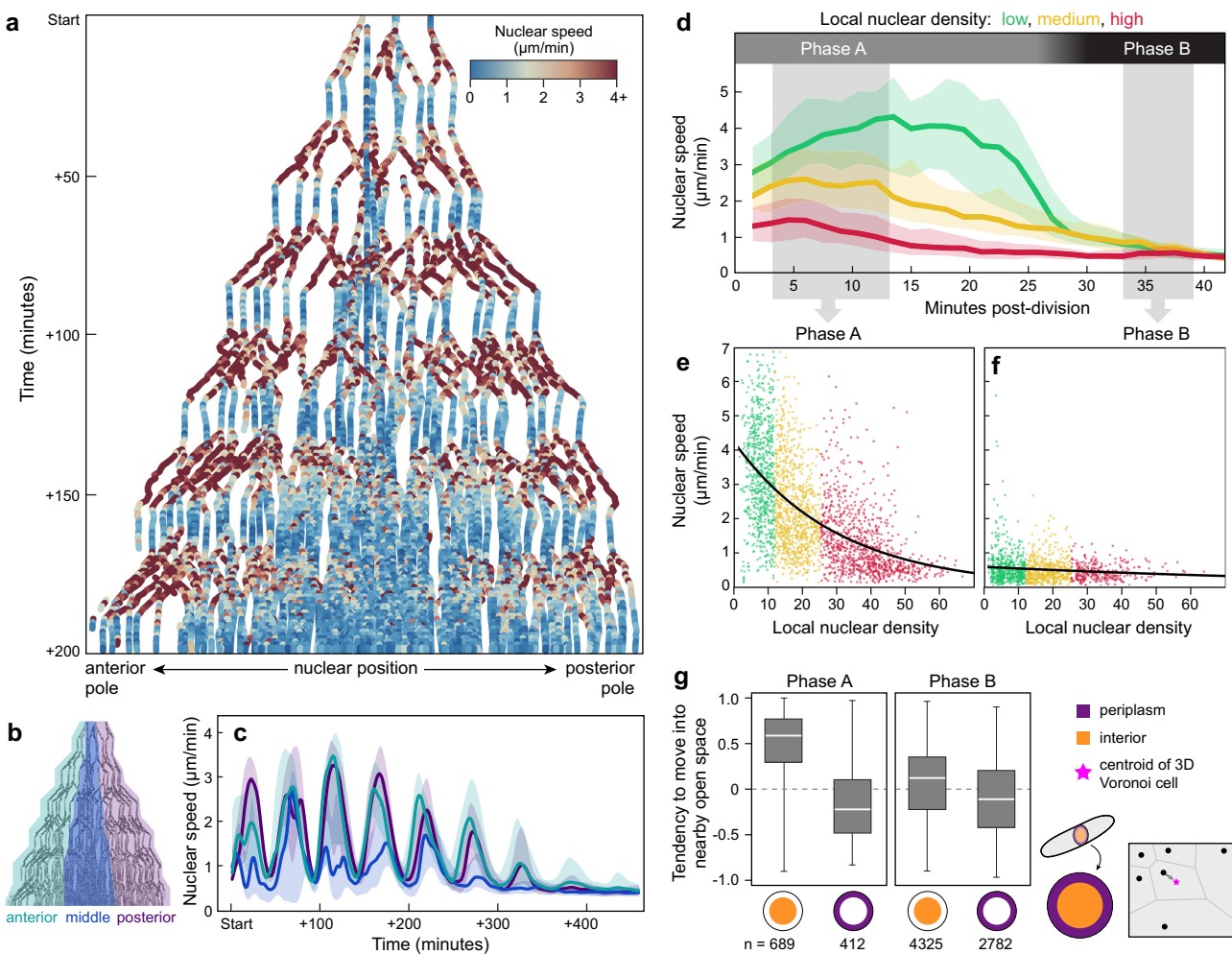

**Fig. 3 After each division, nuclear speed covaries with local nuclear density, and nuclei move into nearby open space. a** Nuclear positions along the anterior-posterior axis (*x*) over 200 minutes of syncytial development (*y*). Each dot represents a nucleus-timepoint, colored by its speed. The first zygotic division occurred ~60% from the anterior pole. Imaging began after the second division. **b** Schematic of nuclei, partitioned into anterior, middle, and posterior thirds (turquoise, dark blue, and purple, respectively). **c** Speed oscillations dissipate earliest in the middle third, where local nuclear density is highest. Center line represents median, shaded regions represent $25^{th}$–$75^{th}$ percentiles. **d–f** Nuclei from all time points and positions were grouped into bins, according to local density (number of nuclei within a 150-µm radius; "low": < 11 nuclei, "medium": $\geq$ 11 and < 29 nuclei, "high": $\geq$ 29 nuclei). **d** Nuclear speed traces after each mitosis concluded. Nuclei move relatively quickly after a division and then slow down. We refer to these periods as "Phase A" and "Phase B". Center line represents median, shaded regions represent $25^{th}$–$75^{th}$ percentiles. **e, f** Nuclear speed versus local density. Data are shown from two periods post-division: $t = 3$ to $t = 13.5$ min (**e**) and $t = 33$ min to $t = 39$ min (**f**). Black line is the best-fit curve of the form $y = y_0 e^{-x/x_0}$, yielding a density scale ($x_0$) of 30.4 density units for Phase A (90% CI: 28.8-32.1) and 120.4 density units for Phase B (90% CI: 73.4-167.4). **g** We calculated a nucleus's tendency to move into nearby open space as its movement vector's correlation with the vector from its current position to the centroid of its Voronoi cell (pink star). Nuclei were also subdivided into those in the "interior" of the embryo, or in the periplasm (defined here as within 75 µm of the eggshell. The distribution of single time-step nuclear movements (n) is shown for each bin. Phase A nuclei in the interior tend to move into nearby open space, but not when they are in the periplasm (left). Phase B nuclei tend not to do so regardless of where they are (right). White line represents median, boxes represent $25^{th}$–$75^{th}$ percentiles, whiskers show range. Source data are provided as a Source Data file.

associated with local density. We concluded that nuclear speeds covaried with local nuclear density and time in the cell cycle.

**In the cytoplasm, nuclei tend to move into nearby unoccupied space.** Nuclear speed alone cannot achieve axial expansion: nuclei also need to move in appropriate directions to ensure even distribution of nuclei in the blastoderm. The directionality cannot be uniform for all nuclei, however, otherwise all nuclei would end up crowded together in a single region of the embryo. To investigate what might predict directionality, we began by observing the global features of nuclear paths during syncytial blastoderm formation. Once syncytial nuclei reach the periplasm, they remain there until cellularization occurs[13,34] (Fig. 1a, b), suggesting that

*G. bimaculatus* nuclei, like *D. melanogaster* nuclei, ultimately become anchored in the periplasm[15,42–45]. Compared to the synchronously emerging nuclei of *D. melanogaster*[15], however, *G. bimaculatus* nuclei move along trajectories that are varied, with some reaching the periplasm as many as 240 minutes later than others (Fig. 1a, b). We hypothesized that as *G. bimaculatus* nuclei move through the cytoplasm, they preferentially move into nearby unoccupied space, which might explain the diversity of traveled paths.

To test this hypothesis, we binned nuclei into those that are in the periplasm and those that are not (Fig. 3g, schematized in orange and purple at right). Then, for the nuclei in each subset we calculated a "space-seeking score", defined as the correlation of a nucleus's instantaneous movement vector with the vector that is

oriented toward the most open space relative to the positions of its neighbors in 3D space (schematized in Fig. 3g, bottom right; see details of this calculation in the Supplemental Information). We found that when they are not in the periplasm, Phase A nuclei tend to move into nearby open space (median space-seeking score = 0.61; Fig. 3g), whereas Phase B nuclei do not (median space-seeking score = 0.12; Fig. 3g). Once nuclei reach the periplasm, however, they do not tend to move into open space during either Phase A or Phase B (median space-seeking score = -0.21 and -0.10, respectively; Fig. 3g). Consequently, they remain in the periplasm once they arrive there, forming the blastoderm, rather than moving back into the central yolk mass. We concluded that except for when they are in the periplasm, Phase A nuclei move preferentially into nearby unoccupied space.

**A simulation framework of cricket syncytial development**. We asked whether the nuclear movements during *G. bimaculatus* blastoderm formation could be explained by models that had been previously proposed for nuclear movement in other contexts. We considered three such candidate models of nuclear movement: (1) Cytoplasmic flows that move nuclei[17,18]; (2) Mutual, repulsive, active forces among nuclei[20,46]; (3) Local, asymmetric, active pulling forces on each nucleus[19,37–40]. This modeling effort was an attempt to assess the relative plausibility of broad bins of explanations for the forces that generate nuclear movements, which could serve as a guide to future molecular-scale empirical work.

These modes of movement are not necessarily mutually exclusive[13], yet for the sake of computational tractability, we assessed each of them in turn. The results presented in Fig. 1c and d contradicted the cytoplasmic flow model (1), so we did not consider it further. Similarly, we found that the empirical data contradicted the mutual repulsion model (2). Specifically, in models with mutual repulsion of nuclei, the magnitude of repulsion is highest when nuclei are closest to one another, and it attenuates with increased distance[25,39]. If a model assumes that a nucleus's speed is directly related to the magnitude of the net force it experiences, a nucleus should move at its lowest speed when it is at the lowest density, i.e. when it is maximally distant from other nuclei. This is the opposite of what we observed in *G. bimaculatus* (Fig. 3e). In *D. melanogaster*, however, recent work on the spacing among nuclei at the syncytial blastoderm stage led to models of mutual nuclear repulsion with symmetric, mutual pushing forces[25,47]. To further evaluate the possibility of model (2), mutual nuclear repulsion in forming the *G. bimaculatus* blastoderm, we selected the model of Dutta and colleagues[25], adapted it to 3D, and implemented it in a simulation framework of preblastoderm syncytial nuclear movements in *G. bimaculatus*. We found that it did not produce a negative relationship between nuclear density and speed at any of the parameter settings we assessed (see Supplementary Note 5.1). We also implemented a new, asymmetric pushing model, which likewise generated a pattern of nucleus movements that was dramatically different from those of real embryos (see Supplementary Note 5.2). Therefore, we did not consider pushing models further, and instead developed (3), a simplified geometric model of a local, asymmetric, active pulling force on each nucleus. Below, we summarize the key features and assumptions of the model.

Our model was inspired by research on nuclear movements in the red bread mold *Neurospora crassa*, the filamentous fungus *Aspergillus nidulans*, the nematode worm *Caenorhabditis elegans*, and *D. melanogaster*, which presented evidence for pulling forces on astral MTs and the microtubule-organizing center (MTOC) associated with each nucleus[19,37,38,40,48,49]. In the present study,

without specific empirical data on the cytoskeletal structures in the *G. bimaculatus* embryo, we abstracted the MT aster and MTOC contributions to the model as a "cloud" and "cloud origin", respectively (schematized in Fig. 4a). We posited that, as in other systems[37,48,50,51], cytoplasmic dyneins or functionally analogous molecules exert pulling forces on the astral MTs. We considered the dyneins to be uniformly distributed in the cytoplasm, and implemented their pulling forces as tugs on each voxel on the surface of each cloud. We divided each tug by a factor of $R^2$ where $R$ represented the distance from the voxel to its cloud origin. Scaling the relative strengths of the forces in this manner is analogous to a set of rods emanating from a point on the nucleus, normalized such that the total pull on a rod was proportional to its length, similar to previous work on *C. elegans* nucleus movement[52]. The sum of all the forces on a cloud origin causes nuclei to move through space and/or rotate. Larger and more asymmetric clouds therefore apply a relatively larger net pulling force (schematized in Fig. 4a). The clouds regrow after each division, but their growth is occluded by one another and by the internal surface of the simulated eggshell (Fig. 4b). If simulated nuclei proliferate in a limited volume, local density increases, which means that over time, clouds grow smaller (Fig. 4c, d). Each nucleus cycles from a Phase A (faster) state to a Phase B (slower) state, then it divides and returns to Phase A. During Phase A, the nucleus's cloud grows to its maximum size (unless spatially constrained) and exerts pulling forces on the nucleus. During Phase B, the cloud is absent.

Here, we briefly describe the full set of inputs to the model; for further details of its computational implementation see Section 4 of the SI. The model was run in a bounded volume that had the same morphology as a real embryo (Supplementary Note 4.1.1). All mitotic division orientations were random (Supplementary Note 4.1.2). Each nucleus was assigned a cycle length based on its local nuclear density. These cycle lengths were drawn from a distribution that was fitted to the empirical density-cycle length relationship (Supplementary Note 4.1.3). The maximum radius of each pulling cloud was set to 150 μm; this is the same size as the local neighborhood that we used to detect the empirical associations between nuclear density and cycle length (Fig. 2c–f), as well as between nuclear density and Phase A speed (Fig. 3e). To assign a magnitude to the pulling forces exerted by the clouds, we calibrated it by fitting a constant factor so that the maximum empirical speed of simulated nuclei matched that of the nuclei in a real embryo (Supplementary Note 4.1.4). We also found that in real embryos, nuclei traveled varied paths with most of them ultimately reaching the periplasm and staying there. It is unknown what causes this periphery-bias in *G. bimaculatus*. In the model, we represented this bias as a small, constant component of each nucleus's movement vector at each time step, tuning its magnitude to match the rate at which real nuclei moved towards the periplasm in empirical data (Supplementary Note 4.1.5).

The model did not include terms for fluid flows, nor maternally provided signals in the yolk, nor any direct mechanical interaction between the force-generating regions around nuclei, each of which is a relevant parameter regulating axial expansion or patterning in the *D. melanogaster* embryonic syncytium[18,27,53]. As in previous work modeling nuclear movements in a syncytial blastoderm[25], we worked in the overdamped limit where viscosity is so large that force was proportional to velocity (see Supplementary Note 4.1 for details). In summary: the model was designed to implement only *local* and *density-associated* effects on nuclear behavior. None of the fitted parameters change as a function of spatial position within the embryo, nor do they change as a function of developmental time. We used this model to determine whether this set of local interactions was sufficient

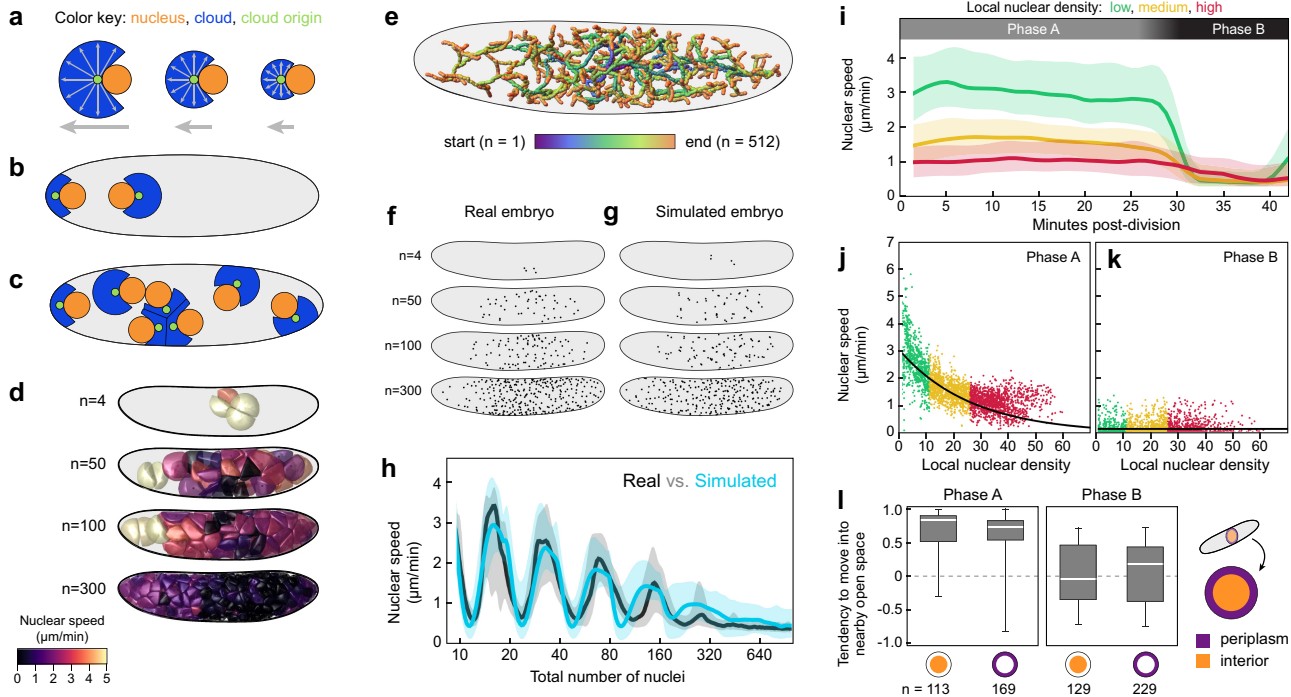

**Fig. 4 A simple model based on local pulling clouds recapitulates many features of _G. bimaculatus_ syncytial development. a–c** Method for simulating syncytial development, schematized in 2D (not to scale). **a** Each nucleus (orange) moves due to a pull from a "cloud" (blue) that grows from a "cloud origin" (green) on the nucleus. Each voxel within a cloud pulls on the nucleus via its origin point (small arrows). The nucleus occludes symmetrical growth of the cloud; thus, the cloud is asymmetric and the net pull on the nucleus is determined by the cloud's size and shape (large arrows). **b** Before a division, two cloud origins are assigned random positions opposite one another on the nuclear surface. Then the nucleus divides. Each daughter nucleus inherits a cloud origin, which grows a new cloud. The eggshell occludes the growth of clouds. **c** Nuclei proliferate and spread. Growing clouds occlude one another; therefore, as local nuclear density increases, clouds are unable to grow as large. Consequently, nuclear speeds are lower in regions with higher nuclear density. The model also includes a general bias in movement toward the periplasm (not depicted). **d** Simulated clouds, colored by their nuclear speeds. **e** 3D paths of simulated nuclei. **f, g** z-projections of nuclear positions in example real (**f**) and simulated (**g**) embryos, stage-matched by total number of nuclei (n). **h** Whole-embryo nuclear speed (y) vs. total nuclear number (x, log scale) from a simulated (turquoise) and real (black) embryo (line represents median, shaded region represents 25th–75th percentiles). **i–l** Simulated nuclear speeds and directions co-vary with density and cycle phase, similar to empirical patterns (see Fig. 3d–g). **i** Simulated nuclear speed traces after division. **j, k** Nuclear speed vs. local density. Best-fit density scale ($x_0$) was 26.8 density units for Phase A (90% CI: 24.2–29.3) and 12,108 density units for Phase B (90% CI: 11,124–13,092). **l** Simulated single time-step nuclear movements (n). Simulated Phase A nuclei in the interior and periplasm tend to move into nearby open space (left). Simulated Phase B nuclei tend not to do so regardless of where they are (right). White line represents median, boxes represent 25th–75th percentiles, whiskers show range. Source data are provided as a Source Data file.

to recapitulate the spatiotemporal distributions of nuclei in wild-type and experimentally altered embryos.

**Simulation results**. First, we wanted to determine whether our implementation of local pulling clouds could generate a density-speed relationship and space-seeking behavior like those of real embryos. As in Fig. 3e and f, we fitted a curve to the speed vs. nuclear density relationship, which yielded density scales for Phases A (faster) and B (slower) of 26.8 density units (90% CI: 24.2 to 29.3) and 12,108 density units (90% CI: 11,124 to 13,092), respectively. Nuclei in the simulated embryo, like those in real embryos, exhibit a negative relationship between density and speed in Phase A (Fig. 4j). In Phase B, with no pulling clouds, nuclear speed does not vary with density at all (Fig. 4k). We also assessed the tendency of nuclei to move into unoccupied space. As in real embryos, simulated nuclei that are not in the periplasm tend to move into unoccupied space during Phase A (median space-seeking score = 0.86) but not during Phase B (median space-seeking score = -0.04; Fig. 4k). However, once simulated nuclei are in the periplasm, unlike in real embryos, simulated nuclei still tend to move into nearby open space during Phase A (median space-seeking score = 0.75; compare Fig. 4k to Fig. 3f). In _D. melanogaster_ there is a cytoskeletal mechanism that holds

nuclei in place once they reach the periplasm[15,42–45]. We suggest that there is likely a similar mechanism at work in _G. bimaculatus_ that has not been included in our model.

Next, we wanted to determine whether the spatial and temporal patterns of _G. bimaculatus_ nuclear movements and positions could be recapitulated solely by the local interactions included in our pulling model. Indeed, we found that although the model did not include any whole-embryo global spatial information, it nevertheless recapitulated the overall spatial distribution of nuclei along the anterior-posterior axis throughout the course of syncytial development (Fig. 4f, g). Similarly, we found that the geometric arrangement of nuclei of the simulated embryo matched that of the real embryo (Fig. 4f, g). In the pulling simulation, the existence of speed oscillations, and an overall tendency for nuclei in higher densities to have longer cell cycles, both result directly from the way the model was fitted to the empirical data. Thus, neither of those features should be viewed as confirmatory results from the modeling. However, although nothing in the model prescribed the decay in speed peaks over developmental time, nuclei in simulated embryos recapitulated the empirically observed time course of speed peak decay (Fig. 4h). This means that the density-dependent mitotic cycle length relationship, calibrated from empirical nucleus-level data,

was sufficient to reproduce the spatiotemporal nuclear proliferation patterns across entire the embryo. In light of the simulation results, we concluded that a local, asymmetric, active pulling force on each nucleus is consistent with most of the observed nuclear behaviors in *G. bimaculatus*.

**Constricting embryos**. Based on the empirical data and simulation results, we hypothesized that the negative relationship between nuclear density and speed emerges only as a consequence of local spatial constraints generated by an asymmetric pulling mechanism. With the descriptive data and simulations alone, however, we could not rule out the possibility that the observed changes in nuclear speed were caused by a spatially localized or temporally varying signal in the cytoplasm. Therefore, to test our hypothesis and assess the alternatives, we experimentally altered the density of nuclei, accomplished by physically manipulating the geometry of embryos. We designed and built a device to constrict an embryo from the outside by wrapping a human hair around it and incrementally increasing the tension on the hair. Specifically, we constricted *G. bimaculatus* embryos width-wise at the beginning of syncytial development, and then mounted them in a glass bottom dish for epifluorescence microscopy (Fig. 5a; see Supplementary Note 6 for detailed methods and Supplementary Software 1 for the design file of the device's components). With this mode of microscopy, we collected $2D + T$ datasets, imaging nuclei through approximately one-third of the $z$ depth of the embryo. This imaging modality necessitated 2D measures of speed and density. Those measures and the total nucleus counts were not comparable on an absolute scale to the equivalent values in the $3D + T$ datasets. Thus, for these analyses, we compared between and within embryos that were all imaged using epifluorescence microscopy. The first zygotic nuclear division typically occurs approximately 60% from the anterior pole along the length of the embryo. We constricted embryos at a position 25% to 35% from the anterior pole (Fig. 5a). By pinching a region down to approximately one-third of the radius of the embryo, we generated two pseudo-compartments in the embryo, each of which was smaller than an unconstricted embryo (Fig. 5b). In both pseudo-compartments of the constricted embryos, the patterns of nuclear density over time and space differed markedly from those of unmanipulated embryos, which enabled us to decouple nuclear speed and density from any as-yet undetected spatially localized cytoplasmic determinants.

First, we compared nuclear behavior between posteriors of constricted embryos to unconstricted embryos. In effect, this allowed us to control for A-P position and developmental time while changing nuclear density. To compare the data from multiple embryos on a single plot, we stage-matched datasets ($n = 3$ embryos per treatment) by the total number of nuclei in each embryo as a proxy for developmental time. As nuclei divide and move within a constricted posterior volume, the total space available to them is reduced compared to an unmanipulated embryo, which causes them to experience higher densities earlier in development than they would otherwise (Fig. 5c). For instance, when there were 200 total nuclei, median density was 20.1 density units in constricted embryos (25th percentile = 18.1, 75th percentile = 22.2) and 15.2 density units in unconstricted embryos (25th percentile = 11.0, 75th percentile = 19.5). The higher density nuclei in constricted embryos moved more slowly than those in stage-matched unmanipulated embryos where the density was lower (Fig. 5d). At 200 total nuclei, median speed was 0.68 µm per minute in constricted embryos (25th percentile = 0.57, 75th percentile = 0.73) and 0.93 µm per minute in unconstricted embryos (25th percentile = 0.85, 75th percentile

= 1.06). Last, we computed all instantaneous nuclear speeds and densities in each constricted posterior and unconstricted dataset, finding that nuclei follow the same density-speed relationship in constricted and unconstricted embryos (Fig. 5e). These results are consistent with a mechanism in which local geometric constraints produce a negative relationship between nuclear speed and density, rather than nuclear speed responding to anteroposterior position or developmental time.

The constricted embryos provided the opportunity to analyze nuclei not only under ectopically high local nuclear densities, but also under ectopically low densities. The latter becomes possible when, in constricted embryos, some nuclei traverse the constricted region and populate the formerly unoccupied pseudo-compartment (Fig. 5a). These nuclei move into a low density region from a comparatively high density region, creating an abrupt change in local density for a subset of nuclei. With this experiment, we altered the developmental time-course of densities that the nuclei experienced. We quantified the nuclear speeds under these conditions, and plotted nuclear speed vs. time for the posterior (ectopically high density) and anterior (ectopically low density) subsets of nuclei (Fig. 5f). We observed that posterior nuclei undergo speed oscillations and overall decreasing speeds, similar to nuclei in unperturbed embryos. We also observed that 50 to 80 minutes after the start of each dataset, nuclei emerge from the constriction (indicated by dotted lines; Fig. 5f), whereupon they speed up as they populate the unoccupied space, slowing down again once local nuclear density increases, ultimately coming to match the speeds of the nuclei in the posterior (compare magenta to orange speed traces; Fig. 5f). We interpreted these results as further evidence of a local mechanism that causes nuclear speed to co-vary negatively with density, which functions independently of developmental time and spatial location.

**Simulating blastoderm formation for embryos with other shapes**. We wanted to know whether our model, parameterized and validated on unmanipulated *G. bimaculatus*, could successfully predict nuclear behaviors in a constricted embryo. We used the same simulation procedure described above, changing only the geometry of the embryo to a constricted shape (Fig. 5g, h). The simulated pulling clouds completely fill the posterior compartment before any nuclei emerge through the constriction (Fig. 5g). This results in a distribution of nuclei over time in the simulations that is qualitatively similar to that of the real constricted embryos (Fig. 5h; compare to 5a). As in the real embryos, the simulated nuclei also exhibit speed oscillations that get slower and with smaller amplitude over time in each pseudo-compartment, but that pattern is offset in time and space for nuclei in the anterior, which abruptly speed up once they pass through the constriction (Fig. 5h). We concluded that simulations broadly recapitulated the experimental results from an atypical embryo geometry, and interpreted this as further support for a model of local, asymmetric, active pulling forces on *G. bimaculatus* syncytial nuclei.

With the aim of uncovering mechanistic principles that might extend beyond this specific case, we asked whether this predictive model of nuclear behavior, derived from observations on *G. bimaculatus*, could be generalized to describe axial expansion in the syncytial embryos of other insects. As a first step in this direction, we asked how well the parameterized model would perform at simulating blastoderm formation in insect species that are either more closely or distantly related to *G. bimaculatus*. The locust *Schistocerca gregaria* is an orthopteran insect, like *G. bimaculatus*, but lays eggs that are 2.5 times longer and 1.5 times wider than *G. bimaculatus* eggs[33,54]. The first nuclear division in

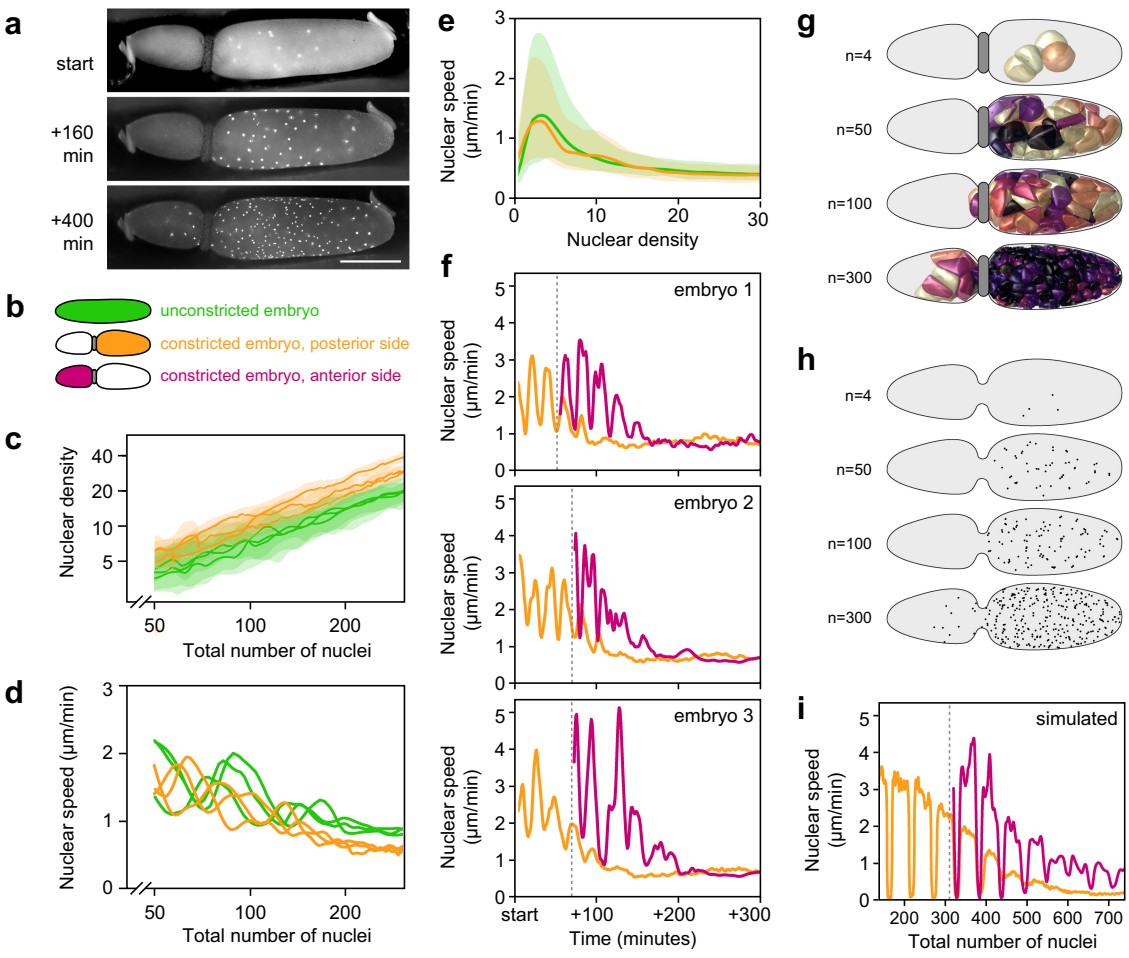

**Fig. 5 Embryo constrictions demonstrate that nuclear density, rather than spatially or temporally localized signals, determines nuclear speed. a** Three time points of an embryo, constricted with a human hair (see Supplemental Note 6), imaged with epifluorescence microscopy. Scale bar represents 500 μm. **b** Color key and schematic for subsets of nuclei represented in this figure. **c** Nuclear density (*y*, log scale) plotted against total number of nuclei (*x*, log scale) for unconstricted embryos (green; *n*=3) and the posterior side of constricted embryos (orange; *n*=3). Line represents median, shaded regions represent 25th−75th percentiles of each embryo's data. Constrictions caused higher nuclear densities for a given number of total nuclei. **d** Median nuclear speed (*y*) vs. total number of nuclei (*x*, log scale) for the same datasets as in **c**. As the posterior sides of constricted embryos fill with nuclei, local densities increase and nuclear speeds decrease, as compared to unconstricted embryos. **e** Nuclear speed (*y*) vs. density (*x*) for all movements in the datasets shown in **c** and **d**. Embryonic constrictions do not change the relationship between speed and density. Center line represents median, shaded regions represent 25th−75th percentiles. **f** Median nuclear speed (*y*) over time (*x*) for three example constricted embryos. Posterior and anterior nuclei are shown in orange and magenta, respectively. Vertical dotted line indicates the timepoint when nuclei first emerge from the constriction into the anterior side. When the nuclei emerge, they speed up as they populate the unoccupied compartment of the embryo, slowing again once local nuclear density increases. **g–i** Our model of pulling clouds (see Fig. 4) qualitatively recapitulated empirical nuclear behaviors in a simulated constricted embryo. **g** 3D renderings of pulling clouds from four selected time points of a simulation of syncytial blastoderm formation in a constricted embryo. Each cloud is colored according to the speed of its nucleus, following the colormap in Fig. 4d. **h** *z*-projections of nuclear positions in a simulated constricted embryo. **i** Median nuclear speed (*y*) vs. total number of nuclei (*x*) for a simulated constricted embryo, plotted as in **f**. Source data are provided as a Source Data file.

*S. gregaria* also occurs closer to the posterior pole than that of *G. bimaculatus*[54–56]. We deployed the computational model described earlier in the text, leaving all parameters the same except for two changes: (1) the morphology of the embryo was set to be an ellipsoid with the volume and approximate shape of the *S. gregaria* embryo;[54] (2) the position of the first division was set to be 85% from the anterior pole along the length of the embryo, similar to that of *S. gregaria*[54]. Based on comparisons to previous work that used fixed embryo preparations, and acknowledging differences in imaging modality, we observed some similarities and some differences between the features of nuclear positioning in the simulated embryo and in real locust embryos[54] (Fig. 6a, b). One similarity was that nuclei reach the posterior pole of the embryo while the anterior two-thirds of the embryo are still

devoid of nuclei. Another is that nuclei form a gradient in their spacing, with the largest internuclear distances at the anterior of the expanding front of nuclei (compare Fig. 6a and b). A detectable difference, however, is that in the real locust embryo, the region with the highest density of nuclei is at the very posterior tip of the embryo. In the simulated embryo, the region with the highest nuclear density remained instead at the position where the first zygotic division occurred. In fact, this was a general outcome in all of our simulations under this model, regardless of embryo shape or size. This could mean that for species whose region of highest nuclear density is far from the position of the earliest division, our model may not effectively describe nuclear behavior in those species without an additional parameter to create directed nuclear migration. We hypothesize

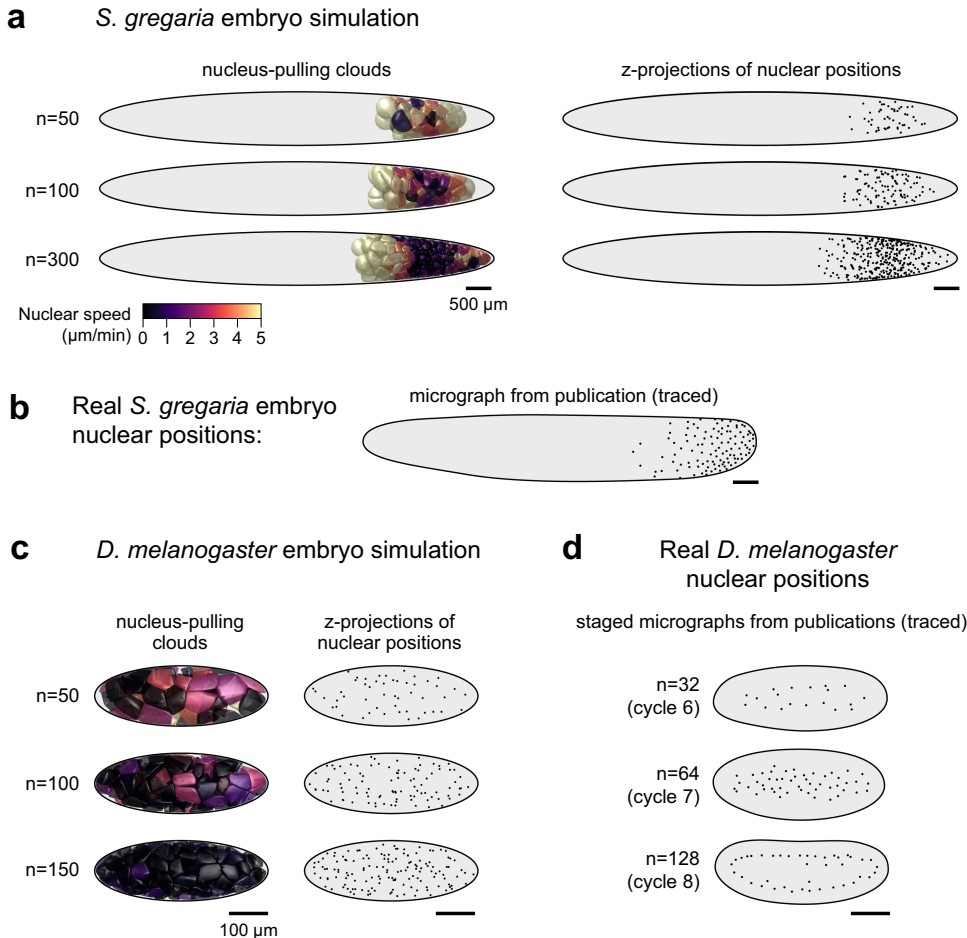

**Fig. 6 The parameterized model of local pulling clouds qualitatively recapitulates blastoderm formation of another orthopteran, but not *D. melanogaster*. a** Selected time points of simulated blastoderm formation in an ellipsoid with the length and width of the locust *Schistocerca gregaria* embryo. The location of the first division was set to 85% from the anterior pole along the length of the embryo, as in *S. gregaria*[54]. Left column: 3D renderings of pulling clouds, colored by simulated nuclear speed. Right column: *z*-projections of simulated nuclear positions. **b** Tracing of a fixed preparation of a *S. gregaria* at 16 hours after egg laying (AEL) at 29 °C and imaged in a manner that captured a subset of the *z* depth of the embryo[54]. **c** Selected time points of simulated blastoderm formation in an ellipsoid with the length and width of the *D. melanogaster* embryo. Left column: 3D renderings of pulling clouds, colored by instantaneous speed. Right column: *z*-projections of simulated nuclear positions. **d** Tracings of published micrographs of *D. melanogaster* embryos from cycle 6, 7, and 8 from Baker and colleagues[20] (cycles 6 and 8) and Deneke and colleagues[18] (cycle 7). In both sources, a subset of the *z* depth of the embryo was imaged[18,20]. For each cycle, the total number of nuclei is shown in parentheses[15].

that a mechanism of local, asymmetric, active pulling forces also operates in the *S. gregaria* preblastoderm embryo, with a possible bias in movement toward the posterior of the embryo.

By contrast, our parameterized model, run with the *D. melanogaster* embryo morphology and first division location, produces arrangements of nuclei that differ qualitatively from those of real *D. melanogaster* embryos. The simulated nuclei spread throughout the volume of the embryo by moving in all directions, with some moving into the periplasm before the rest (Fig. 6c). In real embryos, nuclei spread out predominantly along the A-P axis without moving into the periplasm, then form an ordered shell-like arrangement[18,20] (Fig. 6d), followed by simultaneous movements into the periplasm[20,21] (not depicted in the schematic). We interpret this result as evidence against the hypothesis that there is a shared cellular mechanism that scales with embryo size to generate preblastoderm nuclear behavior across insect taxa[19]. This result is also consistent with recent work on *D. melanogaster*, which demonstrated that cytoplasmic flows generate some of the preblastoderm nuclear movements[18]. In the future it will be fruitful to use experimentally validated computational models to develop hypotheses for nuclear dynamics in other poorly studied systems.

## Discussion

Our computational model was inspired by empirical descriptions of astral MTs and nuclear movements in other contexts[1,19,37,40]. Given the model's effectiveness in capturing *G. bimaculatus* nuclear dynamics, we speculate that cytoplasmic dyneins interacting with astral MTs may indeed be the most likely molecular cause of the asymmetric pulling forces on *G. bimaculatus* syncytial nuclei. There is evidence that such a mechanism may also be present in *D. melanogaster* embryos but obscured by the comparatively dramatic effect of cytoplasmic flows[18]: nuclei in non-flowing cytoplasmic extracts of preblastoderm *D. melanogaster* embryos move apart from one another in a MT- and centrosome-dependent manner that appears to be consistent with a pulling force on asters[19]. If MT-mediated forces are responsible for syncytial nuclear movements, it will be important to know whether the astral MTs are interdigitating and mechanically

interacting[20,27]. Such interactions are absent from our model, but if they were shown to be relevant for *G. bimaculatus*, future modeling work would need to represent the astral MTs in a finer-grained manner than the simple clouds that we described here. Alternatively, asymmetric active forces in *G. bimaculatus* could be generated by a molecular mechanism that does not involve dynein and astral MTs[39]. One possibility is that dynamic remodeling of the actin cytoskeleton in the immediate neighborhood of a nucleus could generate nucleus movements by local, asymmetric fluidization[16]. Another is that asymmetric contractile interactions within an actomyosin network at the periphery of each energid could pull an energid—and the nucleus embedded within it—through the rest of the cytoplasm. It is also possible that MTs and actomyosin contractility jointly contribute to nuclear movements[13,16,57]. For instance, one way that MTs and F-actin can affect nuclear movement was recently shown in *D. melanogaster* blastoderm stage embryos: during a wave of divisions, the ensemble of mitotic spindles drives anisotropic nuclear movement of nuclei, which then return to their original positions in an F-actin-dependent manner[26].

The results of the present study enable us to make inferences about cytoplasmic signals that may regulate the cell cycle behaviors of nuclei during *G. bimaculatus* syncytial development. Similar to *D. melanogaster*[19], *G. bimaculatus* nuclei speed up after each division and then slow down, which suggests that the mechanism driving nuclear movement is coupled to the cell cycle. In *D. melanogaster*, changes in the localization and activity of cytoplasmic Cdk1 and CycB over the course of the cell cycle affect MT and actomyosin dynamics[17,18,58]. We hypothesize that the nucleus-moving mechanism in *G. bimaculatus* is affected in a similar, cell cycle-coupled manner. Unlike *D. melanogaster*[19,21,41,59], however, *G. bimaculatus* cycle duration appears to be locally responsive to nuclear density, rather than coordinated throughout the entire embryo. It has been shown that experimentally increased levels of CycB protein reduce the duration of interphase in *D. melanogaster*[41], and that CycB is degraded at a subcellular spatial scale[60]. Therefore, one possibility is that in *G. bimaculatus*, regions with relatively high nuclear density locally reduce the levels of CycB (or a protein with an analogous function), which causes the duration of interphase to increase accordingly. If true, diffusion would not be predicted to generate uniform cytoplasmic protein levels on relevant time scales in the much larger *G. bimaculatus* embryos. In general, such a mechanism would be consistent with the comparatively heterogeneous cell cycle durations in *G. bimaculatus*. It is also not known how the local environment around each nucleus may change to cause them to stop moving during the latter portion of each cell cycle. During the "slow" phase (Phase B), the MT asters might be absent altogether, or present as a balanced pair on each nucleus[19]. Further descriptive work characterizing MT dynamics in real embryos will be needed to determine which of these best explains the biphasic speed pattern in *G. bimaculatus* nuclei during axial expansion.

Our results have implications for the mechanisms governing subsequent *G. bimaculatus* development. Researchers have begun to describe some of the earliest patterning events in *G. bimaculatus*, including aspects of the establishment of the A-P axis, the dorsal-ventral axis, and the bipartition of the blastodermal cells into embryonic and extraembryonic tissues[34,61–63]. The present study demonstrates that the early *G. bimaculatus* embryo does not have stereotypic divisions nor movements, but instead each nucleus's behavior is determined by the geometry of its neighbors. Collectively, these stochastic interactions, constrained by local rules, generate an emergent uniform spacing of blastodermal nuclei across the embryo. Accordingly, we hypothesize that any maternally provided axial patterning information, if present, is likely to be "stored" in the central cytoplasmic mass or in the periplasm. We also hypothesize that any such putative localized signals will not have a detectable effect on syncytial nuclear behavior. By extension, we predict that the subdivision of the blastoderm into the embryonic anlage and extraembryonic tissues[63] is not determined during the preblastoderm stage, and that the two populations of cells only become separate lineages after a blastoderm has formed.

The computational model we present for blastoderm formation enables us to make predictions about early embryogenesis in other insect species. Insect eggs have a wide range of shapes and sizes[64,65], and the egg determines the shape and size of the syncytial embryo[9,10]. In our simulations, embryo morphology—as well as the location of the initial zygotic division—plays a major role in determining patterns of nuclear movement over the course of blastoderm formation. If an asymmetric pulling mechanism drives syncytial nuclear movements in other insect species, we predict that patterns of syncytial nuclear behavior will tend to co-vary predictably with egg morphology. For instance, in an embryo with a higher aspect ratio[64,65] we would predict a front of low-density nuclei moving at relatively high speeds into unoccupied regions of the embryo, trailed by nuclei arranged in a density gradient (and exhibiting concomitant speeds and cell cycle durations). In a spherical embryo (i.e. an aspect ratio of 1) with a centrally located first zygotic nucleus, we would expect comparatively uniform movements and densities over time and space. In smaller embryos, nuclei would reach an equilibrium spacing more quickly than in larger embryos. These predictions could be tested in a straightforward manner by comparing timed and fixed embryo preparations from insect species that are closely related, yet have embryos of different sizes and/or shapes.

We suggest that elements of the empirical and modeling approaches in the current study could also be fruitfully applied to better understand nuclear behaviors in other multinucleate cell types. In mammalian muscle cells, there are numerous types of nuclear movement, with distinct cellular mechanisms implicated in each one[4,66,67]. Likewise, in filamentous fungi, nuclei are moved by several mechanisms, with important roles for cytoplasmic dynein and astral MTs[3,68]. Arbuscular mycorrhizal fungi have multinucleate spores with hundreds of nuclei, organized in a shared volume, but the cellular mechanisms underlying this arrangement are not known[69,70]. It will be illuminating to discover the ways in which MT- and F-actin-based molecular mechanisms are deployed in different distantly related species to generate forces on nuclei, moving them into the proper arrangement at the right time and place. Comparing such mechanisms across taxa will provide insight into how the conserved eukaryotic cellular machinery shapes developmental evolution.

## Methods

**Transgenesis and animal culture**. *G. bimaculatus* cultures were maintained at 28.5 °C in plastic cages, fed dry cat food (Purina Kitten Chow), and supplied with wet cotton in plastic tubes for hydration[33]. We used an established nucleus-marked transgenic line of *G. bimaculatus*[34], in which the endogenous *actin* promoter drives expression of the *G. bimaculatus* Histone 2B (H2B) protein fused to Enhanced Green Fluorescent Protein (EGFP) (this transgenic line is abbreviated hereafter as Act-H2B-EGFP). To label cytoplasm and nuclei together, we generated a new transgenic insertion of a myristoylated and palmitoylated tandem dimer Tomato fluorescent protein (hereafter: mtdT) expressed under the control of the *G. bimaculatus actin* promoter. The mtdT-3xHA sequence was obtained from pUAStattB-mtdT-3XHA[71] (Addgene Plasmid #24355), and cloned into the pXL[Gbact-GFP-pA] vector[34] in place of GFP, to create pXL[Gbact-mtdT-3xHA-pA]. We used this plasmid to generate a stable transgenic line of *G. bimaculatus* by co-injecting it with a plasmid containing the piggyBac transposase coding sequence, and then screening embryos in the next generation for stable transgenic expression[34]. We crossed the Act-mtdT and Act-H2B-EGFP lines to obtain mature F1 females with both transgenes, assessed by using a fluorescence dissection microscope to check for red and green fluorescent protein expression in late embryogenesis. These F1 females with both

transgenes matured to adulthood, and then were crossed to wild-type males. F2 eggs were collected for live imaging. Thus, when imaging the mtdT and H2B-EGFP transgenic proteins in the same embryo, both were maternally provided to the embryos.

**Collecting and culturing embryos.** To collect embryos for live imaging, females were allowed to lay eggs in damp sand for two hours at a time, and then the eggs were separated from the sand with a sieve[33]. Embryos were examined on a fluorescence dissection microscope within five hours of collection. If there were between 2 and 8 nuclei visible, then embryos were mounted for microscopy as described below under *Microscopy*. After imaging, embryos were placed in a 10 mm diameter plastic petri dish (VWR 25384-342), the bottom of which had been covered with Kimwipes (VWR 21905-026) moistened with distilled water. We incubated the dish at 28.5 °C so that embryos could continue to develop. We checked embryos daily and removed any dead ones. Only datasets from embryos that hatched within 16 days of being laid were used for analysis.

**Microscopy.** For 3D + T lightsheet imaging, we mounted embryos individually in a column of 1% (w/v) low-melt agarose (Bio-Rad 1613112) in distilled water. Suspended in the mounting agarose were 1 μm diameter red fluorescent poly-styrene beads (ThermoFisher F8821) at 0.015% of the stock concentration. Lightsheet imaging was conducted with a Zeiss Z.1 Lightsheet microscope controlled by Zen Black software (Zeiss, 2014-2018), with the agarose column immersed in a bath of distilled water, temperature controlled at 28.5 °C. Embryos were imaged one at a time, positioned with the A-P axis oriented vertically. For each time point, z-stacks were captured at 72 ° or 90 ° increments, rotationally distributed about the long axis of the egg. Data were simultaneously captured with 488 nm and 568 nm lasers at a time interval of 90 seconds, with 100 to 200 optical sections per z-stack. Among lightsheet datasets, z-step size ranged from 4 to 10 μm, depending on the overall size of the field of view needed to capture the embryo. For 3D + T imaging of cytoplasm and nuclei together, we individually mounted embryos in a glass-bottom dish (MatTek P06G-1.5-20-F) in a 20 μL puddle of molten 0.5% (w/v) low-melt agarose in distilled water. Then we covered the immobilized embryos in distilled water and imaged them on a Zeiss LSM 880 confocal microscope at 28.5 °C. For 2D + T imaging of whole embryos, they were mounted in agarose microwells[72] and imaged using epifluorescence on a Zeiss Cell Discoverer microscope with a 5x objective, controlled with Zen Blue (Zeiss, 2015-2018). Epifluorescence datasets were captured as a z-stack at each time point. Embryo constrictions were conducted with a custom device that is described in Supplementary Note 6. A design file of the device's components is included as Supplementary Software 1.

**Image processing and segmentation.** Lightsheet datasets were processed using the Multiview Reconstruction plug-in for Fiji (version 2.0.0-rc-30 to 2.1.0/1.5.3)[73,74]. In epifluorescence datasets, z-slices were combined using the Extended Depth of Focus function (Contrast mode) in Zen Blue (Zeiss, 2015-2018). Confocal datasets were processed in Fiji to generate maximum intensity projections. Nucleus tracks were generated with Ilastik[36] and manually corrected with Fiji plug-in MaMuT (2018 version)[35]. Additional image processing details are included in the Supplementary Note 1.

**Measuring and simulating quantitative features of nuclear behavior.** Data analysis and simulations of nuclear movements were performed using custom scripts written in Mathematica (Wolfram, versions 11 and 12) and Python 3. Python code used functions from the following packages: numpy (version 1.18.1), scipy (version 1.4.1), scikit-fmm (version 2019.1.30). We calculated nuclear speed, local nuclear density, rate of change in number of nuclei, and movement toward nearby unoccupied space. See Supplementary Note 2 for the specifics of these calculations and Supplementary Notes 4 and 5 for a detailed description of nuclear movement simulations.

**Figure preparation.** Micrographs for presentation were processed in Fiji[75]. Figures were generated with Mathematica and assembled with Illustrator (Adobe).

**Reporting summary.** Further information on research design is available in the Nature Research Reporting Summary linked to this article.

## Data availability

The data that support this study are available from the corresponding authors upon reasonable request. For all data that were generated in this study, quantitatively analyzed, and then presented as graphs, see the included source data. Additional tracked nucleus data are available at the GitHub repository for this project: https://github.com/hoffmannjordan/gryllus_nuclear_movements[76]. Animals from the *G. bimaculatus* culture are available for sharing from the corresponding authors upon request, provided that the requestor obtains the necessary permits for the transfer and continued maintenance of the culture (the specific permits vary by jurisdiction). The plasmid for

generating the Act-mtdT transgenic line described in the present study will likewise be available from the corresponding authors upon reasonable request. Source data are provided with this paper.

## Code availability

The aforementioned GitHub repository includes the code used to simulate nuclear movements during blastoderm formation and the code used to convert the tracking output from Ilastik into an XML file that was parsable by MaMuT.

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

## Acknowledgements

This work was supported by funds from the National Science Foundation (NSF award IOS-1257217) and the Howard Hughes Medical Institute to CGE, a US Department of Energy (DOE) Computational Science Graduate Fellowship JH, a National Science Foundation Graduate Training Fellowship to S.D., and the Applied Mathematics Program of the US DOE Office of Advanced Scientific Computing Research under Contract DE-AC02-05CH11231 to CHR. TN was supported by a JSPS Overseas Research Fellowship (Received Number 693) from the Japan Society for the Promotion of Science. We thank the Extavour and Rycroft lab members, the Harvard Center for Biological Imaging, the NSF-Simons Center for Mathematical and Statistical Analysis of Biology at Harvard University, supported by NSF Grant DMS-1764269, and the Harvard Faculty of Arts and Sciences Quantitative Biology Initiative for discussion and support. We thank Julie Theriot for discussion of modeling approaches for nuclear movements and Rakeyah Ahsan for assistance with animal culture.

## Author contributions

S.D. and C.G.E. conceived of the project; S.D. and T.N. performed all transgenic and imaging experiments; S.D. and J.H. analyzed the data; J.H. and C.H.R. designed and implemented all mathematical models in consultation with S.D., T.N. and C.G.E.; all authors contributed to writing of the manuscript.

## Competing interests

The authors declare no competing interests.
