## [Peer Review File · Nature Communications]

Nuclear speed and cycle length co-vary with local density during syncytial blastoderm formation in a cricketReviewers' Comments:

Reviewer #1:

Remarks to the Author:

This manuscript reports on the positioning dynamics of syncytial nuclei in the blastoderm embryo of the cricket *G. bimaculatus*. Nuclear positioning in syncytial embryos is crucial as it defines cell size and cell identity after cellularization. The study of syncytial embryo development in species other than the *D. melanogaster* is important as it may reveal significant differences in the mechanisms in place among insect species.

The experimental data in this manuscript is convincing, of high quality, presented in an adequate manner. The analysis of the data is well done. The embryo constriction experiment is really nice, insightful and well documented.

The model formulation and the mechanistic interpretation (considerations of mechanics and forces) are less convincing. In particular, I do not agree with the central statement of the manuscript, that local density drives movement as it raises a confusion about "cause", "consequence" and "constraints". Below are my explanations why that is.

Major:

1) The authors measured nuclear positions over time and assessed division cycle time and speed. They also calculate local nuclear density for each nucleus and find a positive correlation between cycle time and density, and a negative correlation between speed and density. First, this is correctly interpreted as "association" and "covariance". However, this interpretation is replaced by a deterministic terminology (lines 342, 355). I am not convinced that nuclear density drives (causes) positioning. Rather, (co-)variance with local nuclear density emerges from the dynamics that is determined by an equation of motion (force balance law). This equation should take into account any neighbor interaction and boundary constraints (which the authors have considered geometrically in their model, not mechanically), but also the resistance of moving a nucleus through cytoplasm. How can nuclear density alone be the sole basis (or cause) for movement of a nucleus? It would suggest that similar to particle diffusion (which is based on density or concentration gradients) the force distributing the nuclei emerges from the density gradient (according to principles of statistical physics incl. thermal noise). Mechanically, this is difficult to reconcile given that nuclei are large and the cytoplasm is extremely viscous. The molecular basis set out by the authors is also different; molecular motors generating a drag force on astral microtubules, and thus generating a net pulling force. Hence, more intuitively, the dependence of speed on density is an emerging result of force balance considerations; a key parameter for net force on a nucleus besides the active force generation (cause) is the asymmetry of interactions among all its neighbours (constraints), opposed by viscous force during movement (cause). This will determine direction and speed of motion (consequence), and where they position relative to each other (i.e. density). See also point 2) and 3). In that sense, the title should be changed.

On the other hand, the dependence of cycle time on local nuclear density is interesting and suggests a local biochemical kinetics of mitotic checkpoint components (as mentioned by the authors in the Discussion). This relation determines the reoccurrence of migration, the starting time in the equation of motion of a nucleus.

2) The authors state that the model does not assume terms of viscosity (Line 287). How is this statement not contradicting formula 8 in the supplement? The effective viscous force term is described by γ .

3) In the model, the authors propose a mechanism in which a net pulling force (from dynein motors + cargo moving along astral microtubules and -through drag force from the cytoplasm) moves nuclei within the cytoplasm. This is a perfectly valid model, but the core of this model is the asymmetry of

asters, not the pulling of motors. A pushing model with asymmetry would lead to a net force causing similar dynamics.

4) In the model, a critical issue not addressed adequately is: boundary and neighbor interactions. The authors mention that the “pulling clouds” are occluded by one another and by the internal surface of the eggshell (line 271). As far as I understand, this occlusion only generates geometric constraints on the cloud (and as a consequence a reduction in pulling force). However, it should be considered that clouds of direct neighbors mechanically interact, especially when the pulling model of the authors is based on astral microtubules which interdigitate with those from neighbors (Baker et al 1993, see also Deshpande et al 2019 bioRxiv). The combination of interaction with neighbors – which results in repulsion or interface affinity – and pulling (or pushing) through the cytoplasm as a result of asymmetry, can also explain the directionality and speed of migrating nuclei; where there is no neighbor the repulsion is absent and the net force moving the nucleus is larger and directed towards the empty space. This is then seen as “space-seeking behavior”.

5) In the model, while a nucleus divides and enters a phase of migration (phase A) followed by a phase of almost steady position (phase B), the authors state that in phase B clouds are absent. Does that mean that asters are absent or that symmetry is achieved? These are the only possibilities I see if there are no other force components considered. However, it is unlikely that microtubule asters are absent. More likely, this is assuming the presence of two asters with opposite orientation and balancing each other’s force generation.

6) The necessity in the model calibration of a small but consistent (global) attraction towards the periplasm is mysterious. Wouldn’t such a bias require a signal introducing a bias in active force generation? On the other hand, how is a global attraction different from convection i.e. flow? See also point 8).

7) The model is calibrated using the density-dependent cycle time and density-dependent speed distribution. The authors then claim that the model recapitulates quantitatively the empirical data. Bluntly, I would assume that this approach is self-fulfilling – can the authors clarify better the calibration versus predictions of the model? I was only convinced of some predicting power of the model when recapitulating the dynamics of nuclei passing the constriction site in the embryo ligation experiment.

8) The simulation of nuclear dynamics in the larger embryo of *S. gregaria* – although suggesting similarities – cannot recapitulate nuclear density distribution in the empirical data (Fig 6A,B). While an important test of the model, in my view it failed to qualitatively recapitulate nuclear dynamics because nuclear densities are different even visually. This is critical as density is claimed as the core driver of movement and distribution.

9) In *D. melanogaster*, I wonder how the simulation performs if one assumes an effective diameter of the embryo of roughly half its real diameter. Let’s assume that this species has evolved a nuclear exclusion zone at the embryo cell periphery which then retracts with every division cycle, allowing the nuclei to migrate gradually towards the periplasm during cycle 7-9. In this scenario of exclusion, the embryo volume supporting free nuclei movement is limited to the inner portion only. It is conceivable that a simulation of the dynamics of 6 division cycles in an effectively reduced embryo volume leads to a distribution comparable to the stage 6 embryo shown in Fig 6D. This would be an insightful test of the model, and given the experimental evidence for cytoplasmic flow transporting nuclei during cycles 4-6 (Deneke et al. 2019), which the model here presented does not assume necessary.

Minor:

- Line 191: Is it possible to obtain a quantitative measure for how similar speed oscillations are?
- Line 223: Fig. 3H does not exist; I believe the authors means G throughout this section

- Line 269: How large is the max. cloud size relative to average neighbor distance or in comparison with the neighbor distance distribution?
- Line 430: remove one of the words "stored". I am also not sure of this sentence is meant to read like this.

Reviewer #2:

Remarks to the Author:

The manuscript entitled "Local density determines nuclear movements during syncytial blastoderm formation in a cricket" tracks dynamics of nuclei during syncytial development of *G. bimaculatus*.

Quantification suggests that nuclear divisions and movements are correlated with local density which is corroborated with a geometric model based on asymmetric polling forces on nuclei.

The study stands out by the amount and duration of nuclear tracking. This pristine data set forms the basis for the detailed statistical analysis of nuclear dynamics. Sadly, model design and predictions are only connected by geometrical observations and not biological players in the potential mechanism.

Since the core of the manuscript is based on the comparison of *G. bimaculatus* in comparison to *D. melanogaster*, quantification and analysis are strongly influenced by previous studies on *D. melanogaster*.

Therefore, the novelty relies mostly in the pristine tracking data and the model which predicts active pulling forces in contrast to previous knowledge of insect development from *D. melanogaster*.

Since model assumptions currently seem to be very exhaustive, along you get out what you put in, and direct experimental verification of the forces at play is missing if find this claims speculative.

Major concerns:

Repulsive forces organizing nuclei positions have been ruled out on the ground that "Assuming that a nucleus's speed is directly related to the magnitude of the net force it experiences predicts that a nucleus should display its lowest speed when it is at lowest density,... This is the opposite of what we observed..." Obviously the first statement that a nucleus's speed is proportional with the magnitude of the net force is true. Yet, repulsive forces have already been found to lead to surprising nuclei dynamics, see Ref. 27 on *D. melanogaster* for example, here arising from spatiotemporal heterogeneity of repulsive forces. In fact driving nuclei as well into regions of high nuclear density.

The authors find that nuclear dynamics do vary over time post-division. A strong indication that forces acting are also spatiotemporally varying. I understand that knock-outs on the potential MT dynamics might be impossible, but I would love to see attempts on pharmaceutical perturbations or laser cutting that would point experimentally to the forces at play.

Model assumptions are not well presented in the main text. It is not clear which observations are put into the model and which properties are emergent and not a consequence of the model inputs. For example the division cycle distribution is "matched". Does this mean it is already put into the model? I would appreciate if the model were build bit by bit, giving an argument why which input is really necessary and cannot be emergent of the existing model assumptions.

Reviewer #1 (Remarks to the Author):

This manuscript reports on the positioning dynamics of syncytial nuclei in the blastoderm embryo of the cricket *G. bimaculatus*. Nuclear positioning in syncytial embryos is crucial as it defines cell size and cell identity after cellularization. The study of syncytial embryo development in species other than the *D. melanogaster* is important as it may reveal significant differences in the mechanisms in place among insect species.

The experimental data in this manuscript is convincing, of high quality, presented in an adequate manner. The analysis of the data is well done. The embryo constriction experiment is really nice, insightful and well documented.

The model formulation and the mechanistic interpretation (considerations of mechanics and forces) are less convincing. In particular, I do not agree with the central statement of the manuscript, that local density drives movement as it raises a confusion about "cause", "consequence" and "constraints". Below are my explanations why that is.

Major:

The authors measured nuclear positions over time and assessed division cycle time and speed. They also calculate local nuclear density for each nucleus and find a positive correlation between cycle time and density, and a negative correlation between speed and density. First, this is correctly interpreted as "association" and "covariance". However, this interpretation is replaced by a deterministic terminology (lines 342, 355). I am not convinced that nuclear density drives (causes) positioning.

Rather, (co-)variance with local nuclear density emerges from the dynamics that is determined by an equation of motion (force balance law). This equation should take into account any neighbor interaction

and boundary constraints (which the authors have considered geometrically in their model, not mechanically), but also the resistance of moving a nucleus through cytoplasm. How can nuclear density alone be the sole basis (or cause) for movement of a nucleus? It would suggest that similar to particle diffusion (which is based on density or concentration gradients) the force distributing the nuclei emerges from the density gradient (according to principles of statistical physics incl. thermal noise). Mechanically, this is difficult to reconcile given that nuclei are large and the cytoplasm is extremely viscous. The molecular basis set out by the authors is also different; molecular motors generating a drag force on astral microtubules, and thus generating a net pulling force.

OUR RESPONSE: We agree with the reviewer that density should not be considered the sole cause for movement of a nucleus. Rather, our hypothesis is that the relationship of speed and density is best understood as an emergent result of (inferred) force balance considerations, just as the reviewer describes below. We have updated the text of the manuscript to clarify this point in several places (lines 30, 100-101, 238, 347, 369-379, 381-383). We have also revised the specific lines that the reviewer mentioned as having deterministic terminology as follows:

Line 342 in the original manuscript: *“These results are consistent with a mechanism where speed is determined by nuclear density, rather than A-P position or developmental time.”*

In the revised manuscript (lines 417-419) this now reads: *“These results are consistent with a mechanism in which local geometric constraints produce a negative relationship between nuclear speed and density, rather than nuclear speed responding to anteroposterior position or developmental time.”*

Line 355 in the original manuscript: *“We interpreted these results as further evidence that nuclear speed is determined by density, independent of developmental time or spatial location.”*

In the revised manuscript (lines 432-434) this now reads: *“We interpreted these results as further evidence of a local mechanism that causes nuclear speed to co-vary negatively with density, which functions independently of developmental time and spatial location.”*

Hence, more intuitively, the dependence of speed on density is an emerging result of force balance considerations; a key parameter for net force on a nucleus besides the active force generation (cause) is the asymmetry of interactions among all its neighbours (constraints), opposed by viscous force during movement (cause). This will determine direction and speed of motion (consequence), and where they position relative to each other (i.e. density). See also point 2) and 3). In that sense, the title should be changed.

OUR RESPONSE: As mentioned above, we agree with the reviewer that the speed vs. density relationship results from the constraints of neighboring nuclei. We have updated the text in the locations listed above, and also updated the title of the manuscript to avoid any misunderstanding with the phrase “density-dependent”. The updated title is *“Nuclear speed and cycle length co-vary with local density during syncytial blastoderm formation in a cricket”*.

On the other hand, the dependence of cycle time on local nuclear density is interesting and suggests a local biochemical kinetics of mitotic checkpoint components (as mentioned by the authors in the

Discussion). This relation determines the reoccurrence of migration, the starting time in the equation of motion of a nucleus.

OUR RESPONSE: We agree with the reviewer that this an interesting point, and have retained it in the revised manuscript.

2) The authors state that the model does not assume terms of viscosity (Line 287). How is this statement not contradicting formula 8 in the supplement? The effective viscous force term is described by gamma.

OUR RESPONSE: This comment by the reviewer helped us realize that we should rewrite the sentence in line 287 of the original manuscript (lines 344-346 in the revision) to clarify that for our simulations, we worked in the overdamped limit where viscosity is so large that force is proportional to velocity. This is the same approach taken by Dutta et al 2019, where the authors modeled the movement of nuclei in the *D. melanogaster* syncytium.

3) In the model, the authors propose a mechanism in which a net pulling force (from dynein motors + cargo moving along astral microtubules and -through drag force from the cytoplasm) moves nuclei within the cytoplasm. This is a perfectly valid model, but the core of this model is the asymmetry of asters, not the pulling of motors. A pushing model with asymmetry would lead to a net force causing similar dynamics.

OUR RESPONSE: In response to this comment, we have added a new analysis, in which we assessed the predictions from a pushing model with the same asymmetries—indeed, all other parameters except for the force applied by the clouds—as those that we used in the pulling model. The main result is that in the pushing model the nuclei end up having quite different dynamics from those in real embryos, leading to a spatiotemporal distribution of nuclei that is totally different from the pulling model. Overall we found that this pushing model was a much poorer fit to the empirical data than the pulling model. These new results are described in the Results of the main text (lines 300-301), and also summarized in the revised SI in Section 5.2.

4) In the model, a critical issue not addressed adequately is: boundary and neighbor interactions. The authors mention that the “pulling clouds” are occluded by one another and by the internal surface of the eggshell (line 271). As far as I understand, this occlusion only generates geometric constraints on the cloud (and as a consequence a reduction in pulling force). However, it should be considered that clouds of direct neighbors mechanically interact, especially when the pulling model of the authors is based on astral microtubules which interdigitate with those from neighbors (Baker et al 1993, see also Deshpande et al 2019 bioRxiv). The combination of interaction with neighbors – which results in repulsion or interface affinity – and pulling (or pushing) through the cytoplasm as a result of asymmetry, can also explain the directionality and speed of migrating nuclei; where there is no neighbor the repulsion is absent and the net force moving the nucleus is larger and directed towards the empty space. This is then seen as “space-seeking behavior”.

OUR RESPONSE: We agree that the possibility of direct mechanical interaction among neighboring clouds is an important topic for consideration. As the reviewer mentioned, one way that this interaction might manifest is in mutual pushing between adjacent clouds. To address this comment, we therefore assessed this possibility with the new modeling work described in response to point 3 above, and included it in the Results and SI of the revised manuscript. Further work in this direction would need to be grounded in molecular data for *G. bimaculatus* syncytial development that is currently lacking due to technical limitations. We have clarified in

the revised text that our model does not include such mechanical interactions (lines 341-344). We have also added additional material to the Discussion where we describe the hypothesis of cloud-to-cloud and cloud-to-boundary interactions, and suggest it as an avenue for future inquiry (lines 497-501). We also now mention this hypothesis in the SI in Section 5.2.

5) In the model, while a nucleus divides and enters a phase of migration (phase A) followed by a phase of almost steady position (phase B), the authors state that in phase B clouds are absent. Does that mean that asters are absent or that symmetry is achieved? These are the only possibilities I see if there are no other force components considered. However, it is unlikely that microtubule asters are absent. More likely, this is assuming the presence of two asters with opposite orientation and balancing each other's force generation.

OUR RESPONSE: This important point by the reviewer prompted us to include new text in the revised manuscript to address these hypotheses and suggest future experimental work to help distinguish between them. We agree with the reviewer that if MT asters are responsible for the motion of nuclei in cricket syncytial embryos, an important future topic of research should be to understand what happens to these asters, or what other mechanisms may be in place, in the phase of the cell cycle during which the nuclei largely do not move. We agree that the possibilities mentioned by the reviewer are two promising hypotheses, and we have included both of them in the revised manuscript in lines 525-530.

We have also revised the text of the manuscript to clarify that while our modeling work is an effort to assess a number of broad bins of potential mechanistic explanations for nucleus movements (see lines 282-284), we do not have direct evidence for asymmetric astral MTs playing a role in nuclear movement in *G. bimaculatus*. Uncovering what is happening at a molecular level, and then integrating those data into a revised model, will need to be addressed in future studies.

6) The necessity in the model calibration of a small but consistent (global) attraction towards the periplasm is mysterious. Wouldn't such a bias require a signal introducing a bias in active force generation? On the other hand, how is a global attraction different from convection i.e. flow? See also point 8).

OUR RESPONSE: We agree that this global attraction is mysterious, and we agree that a bias in active force generation is a strong candidate for the source of this bias. As we do not currently have empirical evidence in favor of or against this hypothesis, we have revised the text of the SI to say that this would be an important topic for future work (see SI Section 4.2.5).

As for whether a global bias is different from convection/flows: our view is that if there were flows in the cytoplasm, these would be unlikely to produce a signal of a generalized global bias towards the eggshell, because outward flows in some regions would need to be balanced by inward flows elsewhere (this was also shown for *D. melanogaster* in Deneke et al 2019). Moreover, if there were cytoplasmic flows, we would expect that non-sister nuclei that were relatively close to one another would tend to have correlated movements. We found, however, that such nuclei had no correlation in their movements (Fig. 1D).

7) The model is calibrated using the density-dependent cycle time and density-dependent speed distribution. The authors then claim that the model recapitulates quantitatively the empirical data. Bluntly, I would assume that this approach is self-fulfilling – can the authors clarify better the calibration versus

predictions of the model? I was only convinced of some predicting power of the model when recapitulating the dynamics of nuclei passing the constriction site in the embryo ligation experiment.

OUR RESPONSE: This comment from the reviewer helped us realize that we needed to further clarify exactly which quantities were used to calibrate the model and which ones were assessed as predictions. To address this important point, the components of the model are now described in finer detail in lines 323-350 of the main text of the revised manuscript. The predictions and outputs of the model are also now summarized more clearly in the revised manuscript (see lines 352-379). Here we also briefly summarize those predictions for the benefit of the reviewer: In addition to predicting the effect of the constriction on nuclear speeds (Fig. 5G-I), the model recapitulates the empirical data in two additional ways:

(1) The model does not include any whole-embryo global spatial information, but nevertheless recapitulated the overall spatial distribution of the nuclei along the anterior-posterior axis throughout the course of syncytial development (Fig. 4F and G).

(2) There is nothing in the model that describes the *decay* in speed peaks from the whole population of nuclei over time; rather we only calibrated the density-dependent cycle length from nucleus-level data. The model nevertheless recapitulated the time course of speed peak decay (Fig. 4H).

8) The simulation of nuclear dynamics in the larger embryo of *S. gregaria* – although suggesting similarities – cannot recapitulate nuclear density distribution in the empirical data (Fig 6A,B). While an important test of the model, in my view it failed to qualitatively recapitulate nuclear dynamics because nuclear densities are different even visually. This is critical as density is claimed as the core driver of movement and distribution.

OUR RESPONSE: In response to this valid point by the reviewer, we revised the text of the Results section to clarify which features of the real distribution of the *S. gregaria* nuclei are (in our view) qualitatively similar to that of the simulation run in the *S. gregaria* egg geometry, and which features are not. These updates to the text can be found in lines 461-474. That said, we appreciate that the reviewer (and readers) might nonetheless have reasonable disagreements with us over what constitutes a meaningful visual similarity, and that is also a welcome response to this figure.

We wish to clarify that we do not claim that the simulation and empirical data are identical, nor do we conclude that these cricket and locust species employ the same cellular mechanisms to regulate nuclear movements during axial expansion. Rather, our aim was to communicate that the model output, in this case, serves as a quantitative hypothesis. We included this comparison in the manuscript to provide an example of the sort of predictions that our model can make, and to motivate further experimental work in the future.

9) In *D. melanogaster*, I wonder how the simulation performs if one assumes an effective diameter of the embryo of roughly half its real diameter. Let's assume that this species has evolved a nuclear exclusion zone at the embryo cell periphery which then retracts with every division cycle, allowing the nuclei to migrate gradually towards the periplasm during cycle 7-9. In this scenario of exclusion, the embryo volume supporting free nuclei movement is limited to the inner portion only. It is conceivable that a simulation of the dynamics of 6 division cycles in an effectively reduced embryo volume leads to a distribution comparable to the stage 6 embryo shown in Fig 6D. This would be an insightful test of the

model, and given the experimental evidence for cytoplasmic flow transporting nuclei during cycles 4-6 (Deneke et al. 2019), which the model here presented does not assume necessary.

OUR RESPONSE: This comment made us realize that we could be more explicit in explaining our rationale for the model we develop and test here (those revisions are summarized above in responses to previous points). In *G. bimaculatus*, we did not include any flows in our model because we obtained empirical evidence that cytoplasmic flows are not playing a detectable role in moving the nuclei through the syncytium (Fig. 1E). In contrast, to be useful for explaining nuclear movement in *D. melanogaster*, a model would indeed need to incorporate cytoplasmic flows because the experimental evidence for their existence is quite strong (including but not limited to Deneke et al. 2019). Indeed, we included Fig. 6C and D to illustrate the differences between the true *D. melanogaster* nuclear positions and those that our model would produce in an embryo the shape and size of *D. melanogaster*. In other words, we show that differences in nuclear dynamics between the two species cannot be explained solely by egg size and initial nucleus position.

As for a retracting exclusion zone, we agree with the reviewer that it is an interesting idea for understanding *D. melanogaster*, and it would be a worthwhile endeavor for future work to determine whether that formulation is consistent with empirical nuclear movements and pharmacological experiments in that species (e.g. Baker et al 1993). However, we consider that work beyond the scope of the present paper, and thus have not modified the manuscript in response to this comment.

Minor:

- Line 191: Is it possible to obtain a quantitative measure for how similar speed oscillations are?

OUR RESPONSE: To address this question, we have now added a measure of similarity for this comparison: we computed the correlation of the peak times between each of the datasets. The updated sentence containing this quantitative correlation can be found on lines 231-232 of the revised manuscript.

- Line 223: Fig. 3H does not exist; I believe the authors means G throughout this section

OUR RESPONSE: The reviewer is correct. To address this comment we have corrected that panel reference throughout the manuscript (lines 265-275).

- Line 269: How large is the max. cloud size relative to average neighbor distance or in comparison with the neighbor distance distribution?

OUR RESPONSE: We have added information to answer this important question to Section 4 of the revised SI.

When there are ~100 nuclei, neighbor distance is 132 ± 33 microns (mean \pm SD).
When there are ~350 nuclei, neighbor distance is 78 ± 35 microns (mean \pm SD).
When there are ~1200 nuclei, neighbor distance is 40 ± 14 microns (mean \pm SD).

In our simulations, we used a maximum cloud radius of 150 microns. This means that in the earliest portion of syncytial development, some of the nuclei—especially those that are on the expanding “front”—have clouds that reach the maximum radius. Later on, however, virtually all of

the nuclear clouds become occluded by other clouds or by the eggshell boundary before they reach their maximum radius.

- Line 430: remove one of the words “stored”. I am also not sure of this sentence is meant to read like this.

OUR RESPONSE: We thank the reviewer for noticing this inadvertent typographical error. We have revised that sentence for clarity. It now reads: “Accordingly, we hypothesize that any maternally provided axial patterning information, if present, is likely to be “stored” in the central cytoplasmic mass or in the periplasm. We also hypothesize that any such putative localized signals will not have a detectable effect on syncytial nuclear behaviour.” (lines 538-541)

Reviewer #2 (Remarks to the Author):

The manuscript entitled “Local density determines nuclear movements during syncytial blastoderm formation in a cricket” tracks dynamics of nuclei during syncytial development of *G. bimaculatus*. Quantification suggests that nuclear divisions and movements are correlated with local density which is corroborated with a geometric model based on asymmetric pulling forces on nuclei.

The study stands out by the amount and duration of nuclear tracking. This pristine data set forms the basis for the detailed statistical analysis of nuclear dynamics. Sadly, model design and predictions are only connected by geometrical observations and not biological players in the potential mechanism. Since the core of the manuscript is based on the comparison of *G. bimaculatus* in comparison to *D. melanogaster*, quantification and analysis are strongly influenced by previous studies on *D. melanogaster*. Therefore, the novelty relies mostly in the pristine tracking data and the model which predicts active pulling forces in contrast to previous knowledge of insect development from *D. melanogaster*. Since model assumptions currently seem to be very exhaustive, along you get out what you put in, and direct experimental verification of the forces at play is missing if find this claims speculative.

OUR RESPONSE: We are gratified that this reviewer appreciated the novelty and depth of our tracking data. In response to the general comment on the degree of new insight provided by our model, in this revision we have expanded and clarified our modeling work (see above in our responses to Reviewer 1 points 3 and 7), to show that the simulations we perform clearly enable us to distinguish between competing types of putative force generation on nuclei. We also show that the pulling model is not overfit to empirical inputs: rather, it predicts several features of the unmanipulated empirical data that are not included parameters at all (summarized in lines 352-379 of the revision), and it also predicts the results of experimentally altered embryos (lines 436-446 of the revision).

Major concerns:

Repulsive forces organizing nuclei positions have been ruled out on the ground that “Assuming that a nucleus’s speed is directly related to the magnitude of the net force it experiences predicts that a nucleus should display its lowest speed when it is at lowest density,... This is the opposite of what we observed...” Obviously the first statement that a nucleus’s speed is proportional with the magnitude of the net force is true. Yet, repulsive forces have already been found to lead to surprising nuclei dynamics, see Ref. 27 on *D. melanogaster* for example, here arising from spatiotemporal heterogeneity of repulsive forces. In fact driving nuclei as well into regions of high nuclear density. The authors find that nuclear dynamics do vary over time post-division. A strong indication that forces acting are also spatiotemporally

varying. I understand that knock-outs on the potential MT dynamics might be impossible, but I would love to see attempts on pharmaceutical perturbations or laser cutting that would point experimentally to the forces at play.

OUR RESPONSE: To address this point, also raised by Reviewer 1 in their point 3, we have added a new modeling analysis. Specifically: in order to better assess the alternative possibility that repulsive forces are generating nucleus movements, we re-run our simulation with the same model inputs as before, except that the clouds now generate repulsive forces against one another. This produced a completely different spatiotemporal arrangement of nuclei that was a much poorer fit to the empirical data than the pulling model we propose (see Section 5 in the SI and lines 285-304 in the revised main text).

With regards to the suggestion of chemical treatments or laser cutting, we agree that these would be quite informative for illuminating the forces on nuclei. However, those experimental techniques have not yet been established for this animal with the degree of resolution needed to make their results interpretable for the purposes of addressing this point. Thus, such experiments will need to be the subject of future work.

Model assumptions are not well presented in the main text. It is not clear which observations are put into the model and which properties are emergent and not a consequence of the model inputs. For example the division cycle distribution is “matched”. Does this mean it is already put into the model? I would appreciate if the model were build bit by bit, giving an argument why which input is really necessary and cannot be emergent of the existing model assumptions.

OUR RESPONSE: To address this point, also raised by Reviewer 1 in their point 7, we have expanded the section on the model assumptions in the main text to clarify exactly which quantities were put into the model, and which ones emerged. These updates can be found in lines 327-350 of the revised manuscript.

In the specific case of the division cycle distribution, this was an *input* to the model. We measured the mean and standard deviation of cell cycle length at each density bin. Then, for each nucleus in a simulation, we computed its local density at the start of its cell cycle and drew from a distribution of cell cycle lengths with the empirical mean and standard deviation (also described in Section 4.2.3 of the SI).

Reviewers' Comments:

Reviewer #1:

Remarks to the Author:

I appreciate the efforts of the authors to consider and implement the suggestions from my initial review comments. All points raised have been addressed, either by modifications, additional data or by commenting satisfactorily. Although I and the authors have acknowledged that there are open questions still to be addressed (in future work), I am happy with the current version and support publication of this manuscript.

Reviewer #2:

Remarks to the Author:

Reading through the response to the reviewers I am surprised by the limited changes the authors did to incorporate the very detailed concerns regarding their model. To me too many concerns are answered with "we have revised the text of the SI to say that this is would be an important topic for future work" rather than improving their manuscript. The model does not show the rigor to grant my acceptance of its publication.

Given that previous detailed comments did not result in a more rigorous presentation and investigation of the model predictiveness it seems not suitable to add more detailed comments. My gist is that the predictiveness of the model is not sufficient. As the authors summarize in their response to the reviewers, their predictions are limited to a) the visual comparison of the spatial distribution of nuclei and b) the oscillations, precisely the decay in nuclei speeds over time. To me b) is not a prediction but a direct result of alternating phases A and B whose different nuclear speeds are a result of data fits to nuclei speed vs nuclear density and cell cycle versus nuclear density. While a) is a solely visual prediction that to my opinion could also arise due to other interactions, potentially as elaborate as those of Lv et al Curr Biol 2020, which has not been investigated by the authors. Reversing the sign of the forces in the 'cloud' model, as now investigated by the authors, does not provide an accurate model of the repulsive force of the overlapping MT asters of daughter nuclei and so is bound to not accurately describe the nuclei dynamics.

Reviewer #1 (Remarks to the Author):

I appreciate the efforts of the authors to consider and implement the suggestions from my initial review comments. All points raised have been addressed, either by modifications, additional data or by commenting satisfactorily. Although I and the authors have acknowledged that there are open questions still to be addressed (in future work), I am happy with the current version and support publication of this manuscript.

OUR RESPONSE: We are grateful for the reviewer's positive feedback.

Reviewer #2 (Remarks to the Author):

Reading through the response to the reviewers I am surprised by the limited changes the authors did to incorporate the very detailed concerns regarding their model. To me too many concerns are answered with "we have revised the text of the SI to say that this is would be an important topic for future work" rather than improving their manuscript. The model does not show the rigor to grant my acceptance of its publication.

Given that previous detailed comments did not result in a more rigorous presentation and investigation of the model predictiveness it seems not suitable to add more detailed comments. My gist is that the predictiveness of the model is not sufficient. As the authors summarize in their response to the reviewers, their predictions are limited to a) the visual comparison of the spatial distribution of nuclei and b) the oscillations, precisely the decay in nuclei speeds over time. To me b) is not a prediction but a direct result of alternating phases A and B whose different nuclear speeds are a result of data fits to nuclei speed vs nuclear density and cell cycle versus nuclear density. While a) is a solely visual prediction that to my opinion could also arise due to other interactions, potentially as elaborate as those of Lv et al Curr Biol 2020, which has not been investigated by the authors. Reversing the sign of the forces in the 'cloud' model, as now investigated by the authors, does not provide an accurate model of the repulsive force of the overlapping MT asters of daughter nuclei and so is bound to not accurately describe the nuclei dynamics.

OUR RESPONSE: We appreciate the reviewer's desire for a modeling effort that accurately predicts empirical patterns. The reviewer raises valuable points, and we realized that we were not sufficiently clear in explaining our results and interpretation of the modeling in the manuscript. We have updated the manuscript accordingly, and we also explain our approach in more detail below.

First, we want to clarify the role that modeling is intended to play in the present study. Here, our aim for the manuscript is to document pre-blastoderm nuclear movement phenomena for the first time in this species, and then assess broad categories of mechanisms that could underlie them. In our view, in such a paper, the scientific usefulness of a given model is not found in its capacity to perfectly recapitulate an empirical pattern. Rather, in this context, a model serves as a way to ask whether a given set of simple assumptions are sufficient to generate an output state with certain attributes (see lines 453-455 in the manuscript). Each of the models that we present in the paper, including the "pulling" model we favor, as well as the two different "pushing" models, are exercises to help us (and the field as a whole) to assess the consequences of a given set of assumptions, and thereby guide future empirical work toward the most likely molecular mechanisms that are operating. In each of our models, we have fit the same empirical quantities (maximum nucleus speed, overall egg shape, density-to-cycle-length relationship, etc.), changing

only the source of the force on each nucleus. These models of nuclear movement were based on long-standing hypotheses about nuclear positioning in the literature, each of which has empirical support in different contexts.

One key finding of our modeling work is that each of these models produces nuclear behaviors that are completely and qualitatively distinct. We also found that the output of one of the models is a much *closer* fit to the empirical data than the others, but this does not prove that our favored model is capturing the molecular mechanisms that are controlling nuclear movements in reality. There are certainly still more ways to model nuclear movements. For the present manuscript, we have limited our efforts to those models that we view as being most plausible. The reviewer suggests another—something like the yo-yo behavior described in Lv et al., *Current Biology* 2020. In this paper, Lv et al. describe a stage of *Drosophila* development after a syncytial blastoderm (i.e. at a stage later than the one we consider in our work) has already formed, when waves of mitosis sweep across the shell-like arrangement of nuclei in the periplasm. The ensemble of spindles drives anisotropic nuclear movement through a displacement of several nuclear diameters, followed by a return to the original positions. We appreciate this suggestion, but we view this particular alternative as being poorly suited to explaining the dominant nuclear movements in the cricket pre-blastoderm embryo, because during the period of development on which we focus, mitoses are much less synchronous than they are in *Drosophila* (see Fig. 2B). Moreover, the nuclear movements that are responsible for transporting nuclei throughout the anterior-posterior length of the blastoderm span many successive cell cycles, are not restricted to a shell-like arrangement in the periplasm, are non-recurrent, and cover 10s of nuclear diameters in distance. These are all substantial differences from the nuclear positions and movements at the later developmental stage that the Lv et al. model was developed to explain. For those reasons, we view an effort to adapt the Lv et al. model as being beyond the scope of the current study. Nonetheless, we think it is a useful general suggestion for the field, and we have updated the manuscript to include it (see lines 686-689).

Second, on the topic of just how accurately the “pulling” model in our paper recapitulates the empirical data, it is helpful that the reviewer summarized their understanding of our results, because this made it clear that we should explain our findings more clearly. We comment on the separate components of the reviewer’s summary in turn:

“[T]heir predictions are limited to a) the visual comparison of the spatial distribution of nuclei [...] that to my opinion could also arise due to other interactions [...]”

To clarify, we found that the “pulling” model generated a spatial distribution of nuclei that was qualitatively similar to that of the empirical data; moreover, this similarity held *throughout the entirety* of blastoderm formation. We could have quantified this similarity and compared it to alternatives, but we judged that there was little point in doing so when the alternatives produced results that were a strong mismatch to the empirical data in obvious ways (see, for instance, Fig. S2, where the nuclei do not travel the length of the egg at all).

“[...] and b) the oscillations, precisely the decay in nuclei speeds over time. To me b) is not a prediction but a direct result of alternating phases A and B whose different nuclear speeds are a result of data fits to nuclei speed vs nuclear density and cell cycle versus nuclear density.”

The reviewer is correct that the local cycle length-density relationship is fitted to the data. The local *speed*-density relationship is not directly fitted to the data, but rather only the peak speed is

fitted, and then the speed-density relationship emerges from local geometric arrangements of nuclei. Furthermore, these quantities are only fitted to the data at the scale of each nucleus's neighborhood, measured independently of spatial position or developmental time. The whole-embryo pattern of speed peaks over time emerges from how all the nuclei move as a collective, leading to a specific distribution of densities that in turn gives rise to the speed decay pattern. That said, the reviewer makes an important point about fitting to the empirical data, and we want to make sure readers are not misled. Therefore, we have updated the manuscript to be clearer that the existence of oscillations and an overall tendency to lengthening cell cycles both result directly from the way the model was fitted to the empirical data, and therefore neither of those features should be viewed as a confirmatory result from this plot (see lines 543-547).

Finally, and most importantly, we want to point out that there is another major result of the modeling that the reviewer did not include in their summary of the modeling results: the "pulling" model predicted the effect of the experimental constriction on nuclear speeds and positions (Fig. 5G-I). We view this as the strongest support for the pulling model (see lines 620-622).